# Support Regularized Sparse Coding and Its Fast Encoder

**Yingzhen Yang**[1,2]**, Jiahui Yu**[2]**, Pushmeet Kohli**[3]**, Jianchao Yang**[1]**, Thomas S. Huang**[2]

[1] Snap Research
superyyzg@gmail.com,jianchao.yang@snapchat.com
[2] Beckman Institute, University of Illinois at Urbana-Champaign
{jyu79,t-huang1}@illinois.edu
[3] Microsoft Research
pkohli@microsoft.com

## Abstract

Sparse coding represents a signal by a linear combination of only a few atoms of a learned over-complete dictionary. While sparse coding exhibits compelling performance for various machine learning tasks, the process of obtaining sparse code with fixed dictionary is independent for each data point without considering the geometric information and manifold structure of the entire data. We propose Support Regularized Sparse Coding (SRSC) which produces sparse codes that account for the manifold structure of the data by encouraging nearby data in the manifold to choose similar dictionary atoms. In this way, the obtained support regularized sparse codes capture the locally linear structure of the data manifold and enjoy robustness to data noise. We present the optimization algorithm of SRSC with theoretical guarantee for the optimization over the sparse codes. We also propose a feed-forward neural network termed Deep Support Regularized Sparse Coding (Deep-SRSC) as a fast encoder to approximate the sparse codes generated by SRSC. Extensive experimental results demonstrate the effectiveness of SRSC and Deep-SRSC.

## 1 Introduction

The aim of sparse coding is to represent an input vector by a linear combination of a few atoms of a learned dictionary which is usually over-complete, and the coefficients for the atoms are called sparse code. Sparse coding is widely applied in machine learning and signal processing, and sparse code is extensively used as a discriminative and robust feature representation with convincing performance for classification and clustering (Yang et al., 2009; Cheng et al., 2013; Zhang et al., 2013). Suppose the data $X = [\mathbf{x}_1, \mathbf{x}_2, \ldots, \mathbf{x}_n] \in \mathbb{R}^{d \times n}$ lie in the $d$-dimensional Euclidean space $\mathbb{R}^d$, and the dictionary matrix is $\mathbf{D} = [\mathbf{D}^1, \mathbf{D}^2, \ldots, \mathbf{D}^p] \in \mathbb{R}^{d \times p}$ with each $\mathbf{D}^k \in \mathbb{R}^d$ $(k = 1, \ldots, p)$ being an atom of the dictionary, sparse coding method seeks for the linear sparse representation with respect to the dictionary $\mathbf{D}$ for each vector $\mathbf{x} \in X$ by solving the following convex optimization problem:

$$\min_{\mathbf{D}, \mathbf{Z}} \sum_{i=1}^{n} \frac{1}{2} \|\mathbf{x}_i - \mathbf{D} \mathbf{Z}^i\|_2^2 + \lambda \|\mathbf{Z}^i\|_1 \quad s.t. \ \|\mathbf{D}^k\|_2 \leq c_0, k = 1, \ldots, p$$

where $\lambda$ is a weighting parameter for the $\ell^1$-norm of $\mathbf{z}$, and $c_0$ is a positive constant that bounds the $\ell^2$-norm of each dictionary atom. In (Gregor & LeCun, 2010), a feed-forward neural network named Learned Iterative Shrinkage and Thresholding Algorithm (LISTA) is proposed to produce the approximation for sparse coding (1). The architecture of LISTA is illustrated in Figure 1. The LISTA network involves an finite number of stages wherein each stage performs the following operation on the intermediate sparse code:

$$\mathbf{z}^{(k+1)} = h_\theta(\mathbf{W}\mathbf{x} + \mathbf{S}\mathbf{z}^{(k)}), \quad \mathbf{z}^{(0)} = \mathbf{0} \tag{1}$$

This material is based upon work supported by the National Science Foundation under Grant No. 1318971. Any opinions, findings, and conclusions or recommendations expressed in this material are those of the author(s) and do not necessarily reflect the views of the National Science Foundation.

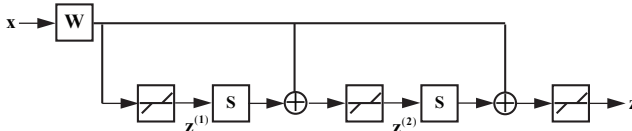

Figure 1: Illustration of LISTA network for approximate sparse coding.

where $h_\theta$ is an element-wise shrinkage function defined as

$$[h_\theta(\mathbf{u})]_k = \text{sign}(\mathbf{u}_k)(|\mathbf{u}_k| - \theta)_+, k = 1, \ldots, p \tag{2}$$

and $(\cdot)_+ = \max\{\cdot, 0\}$ is the positive part of a number. Let $f$ indicate the LISTA network and it generates the approximate sparse code $\mathbf{z} = f(\mathbf{x}, \Theta)$, where $\Theta = (\mathbf{W}, \mathbf{S}, \theta)$ collectively denotes the parameters of the LISTA network. Suppose the optimal sparse codes for the training data $\mathbf{x}_1, \ldots, \mathbf{x}_m$ are $\mathbf{Z}^{*1}, \ldots, \mathbf{Z}^{*m}$, then the parameters $\Theta$ are learned by minimizing the cost function which measures the distance between the predicted approximate sparse codes and the optimal sparse codes: $\mathcal{L}(\Theta) = \frac{1}{m} \sum_{i=1}^{m} \|\mathbf{Z}^{*i} - f(\mathbf{x}_i, \Theta)\|_2^2$. The optimization is performed by stochastic gradient descent and back-propagation. Inspired by LISTA, a series of previous works have designed neural networks to simulate different forms of linear coding and achieve end-to-end training for different tasks such as image super-resolution (Liu et al., 2016) and hashing (Wang et al., 2016).

Sparse coding is widely used to model high-dimensional data. Based on the formulation of sparse coding (1), it can be observed that the sparse code of each data point is obtained independently when the dictionary is fixed, which ignores the geometric information and manifold structure of the high-dimensional data. In order to obtain the sparse codes that account for the geometric information and manifold structure of the data, many regularized sparse coding methods, such as (Liu et al., 2010; He et al., 2011; Zheng et al., 2011; Gao et al., 2013), employ manifold assumption (Belkin et al., 2006). Manifold assumption in these methods imposes local smoothness on the sparse codes of nearby data, namely nearby data are encouraged to have similar sparse codes in the sense of $\ell^2$-distance, and they are termed $\ell^2$-Regularized Sparse Coding ($\ell^2$-RSC). In this paper, we propose Support Regularized Sparse Coding (SRSC). Compared to $\ell^2$-RSC, SRSC captures the locally linear structure of the data manifold by encouraging nearby data to share dictionary atoms. In addition, SRSC enjoys robustness to data noise and preserves freedom in the spare representation of data without constraints on the magnitude of the sparse codes.

The remaining parts of the paper are organized as follows. SRSC and its optimization algorithm, together with $\ell^2$-RSC are introduced in the next section. The theoretical properties of the optimization of SRSC are shown in Section 3, with theoretical guarantee on the obtained sub-optimal solution for each step of the coordinate descent for obtaining the support regularized sparse codes: convergence to the critical point of the objective function and being close to the globally optimal solution. We then show the performance of the SRSC on data clustering, and conclude the paper. We use bold letters for matrices and vectors, and regular lower letter for scalars throughout this paper. The bold letter with superscript indicates the corresponding column of a matrix, and the bold letter with subscript indicates the corresponding element of a matrix or vector. $\| \cdot \|_F$ and $\| \cdot \|_p$ denote the Frobenius norm and the $\ell^p$-norm.

## 2 SUPPORT REGULARIZED SPARSE CODING

### 2.1 CAPTURING LOCALLY LINEAR STRUCTURE: SUPPORT REGULARIZED SPARSE CODING

In this section, we introduce Support Regularized Sparse Coding (SRSC) which is designed to capture the locally linear structure of the data manifold for sparse coding. One of the most important properties of manifold is that it is locally Euclidean, and each data point in the manifold has a neighbourhood that is homeomorphic to a Euclidean space. The success of several manifold learning methods, including LLE (Roweis & Saul, 2000), SMCE (Elhamifar & Vidal, 2011) and Locally Linear Hashing (Irie et al., 2014), is built on exploiting the locally linear structure of manifold. In these methods, the locally linear structure associated with each data point is a linear representation of

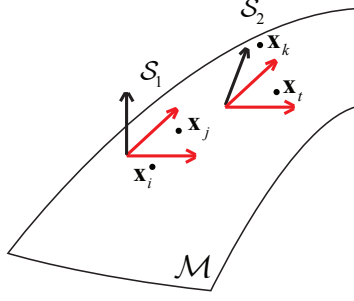

Figure 2: Illustration of capturing the locally linear structure of the data manifold by Support Regularized Sparse Coding. Nearby data are encouraged to share dictionary atoms. In this example, $\mathbf{x}_i$ and $\mathbf{x}_j$ choose three common dictionary atoms so they lie on or close to the local subspace $\mathcal{S}_1$ spanned by the common atoms, and it is the similar case for $\mathbf{x}_t$ and $\mathbf{x}_k$ with local subspace $\mathbf{S}_2$. Due to the smoothness of the support of the sparse codes, neighboring local subspaces, such as $\mathcal{S}_1$ and $\mathcal{S}_2$, can share dictionary atoms. In this example, the two local subspaces share two dictionary atoms marked in red.

that point by a set of its nearest neighbors in a nonparametric manner, from which the low-dimensional embedding complying to the manifold structure of the original data is obtained and used for various learning tasks. In the context of sparse coding, the data lie on or close to the subspaces spanned by the dictionary atoms specified by the nonzero elements of the corresponding sparse codes. Inspired by this observation, we propose to capture the locally linear structure of the data manifold for sparse coding by encouraging nearby data to share the atoms of the dictionary, so that nearby data are on or close to the local subspace spanned by the common dictionary atoms (see Figure 2).

In order to obtain the sparse codes with locally similar support so as to capture the locally linear structure of the data manifold, we propose Support Regularized Sparse Coding (SRSC), which uses support distance to measure the distance between the sparse codes of nearby data. Given a proper symmetric similarity matrix $\mathbf{A}$, the sparse codes $\mathbf{Z}$ that capture the locally linear structure of the manifold minimizes the following support regularization term:

$$\mathbf{R_A}(\mathbf{Z}) = \frac{1}{2} \sum_{i=1}^{n} \sum_{j=1}^{n} \mathbf{A}_{ij} d(\mathbf{Z}^i, \mathbf{Z}^j) \tag{3}$$

$\mathbf{A}$ is usually the adjacency matrix of K-Nearest-Neighbor (KNN) graph, i.e. $\mathbf{A}_{ij} = 1$ if and only if $\mathbf{x}_i$ is among the $K$ nearest neighbors of $\mathbf{x}_j$ or $\mathbf{x}_j$ is among the $K$ nearest neighbors of $\mathbf{x}_i$. Note that KNN is extensively used in the manifold learning literature, such as Locally Linear Embedding (LLE) (Roweis & Saul, 2000), Laplacian Eigenmaps (Belkin & Niyogi, 2003) and Sparse Manifold Clustering and Embedding (SMCE) (Elhamifar & Vidal, 2011), to establish the local neighborhood in the manifold. $d$ indicates the support distance. For two vectors $\mathbf{u}, \mathbf{v}$ of the same size, their support distance is defined below:

$$d(\mathbf{u}, \mathbf{v}) = \sum_{t=1}^{|\mathbf{u}|} (\mathbb{I}_{\mathbf{u}_t=0, \mathbf{v}_t \neq 0} + \mathbb{I}_{\mathbf{u}_t \neq 0, \mathbf{v}_t=0}) \tag{4}$$

where $\mathbb{I}$ is the indicator function. When the support distance between $\mathbf{Z}^i$ and $\mathbf{Z}^j$ is small for nonzero $\mathbf{A}_{ij}$, $\mathbf{x}_i$ and $\mathbf{x}_j$ choose similar atoms of the dictionary for sparse representation. Therefore, SRSC captures the locally linear structure of the data manifold by encouraging nearby data to share dictionary atoms, wherein the common atoms shared by nearby data serve as the basis of the local subspace.

The optimization problem of SRSC is presented below:

$$\min_{\mathbf{D}, \mathbf{Z}} L(\mathbf{D}, \mathbf{Z}) = \sum_{i=1}^{n} \frac{1}{2} \|\mathbf{x}_i - \mathbf{D}\mathbf{Z}^i\|_2^2 + \lambda \|\mathbf{Z}^i\|_1 + \gamma \mathbf{R_A}(\mathbf{Z}) \quad s.t. \ \|\mathbf{D}^k\|_2 \leq 1, k = 1, \ldots, p \tag{5}$$

where $\gamma > 0$ is the weighting parameter for the support regularization term. Similar to (Lee et al., 2006), problem (5) is optimized alternatingly with respect to the dictionary $\mathbf{D}$ and the sparse codes $\mathbf{Z}$ respectively with the other variable fixed.

### 2.1.1 Optimizing with respect to $\mathbf{D}$ with fixed $\mathbf{Z}$

The optimization with respect to $\mathbf{D}$ with fixed $\mathbf{Z}$ is a quadratic programming problem:

$$\min_{\mathbf{D}} \frac{1}{2}\|\mathbf{X} - \mathbf{DZ}\|_F^2 \quad s.t. \ \|\mathbf{D}^k\|_2 \leq 1, k = 1, \ldots, p \tag{6}$$

which can be solved using Lagrangian dual (Lee et al., 2006).

### 2.1.2 Optimizing with respect to $\mathbf{Z}$ with fixed $\mathbf{D}$

We use coordinate descent to optimize (5) with respect to $\mathbf{Z}$ with fixed $\mathbf{D}$:

$$\min_{\mathbf{Z}} \sum_{i=1}^{n} \frac{1}{2}\|\mathbf{x}_i - \mathbf{DZ}^i\|_2^2 + \lambda\|\mathbf{Z}^i\|_1 + \gamma\mathbf{R_A}(\mathbf{Z}) \tag{7}$$

In each step of coordinate descent, the optimization is performed over the $i$-th column of $\mathbf{Z}$, while fixing all the other sparse codes $\{\mathbf{Z}^j\}_{j\neq i}$. For each $1 \leq i \leq n$, the optimization problem for $\mathbf{Z}^i$ is below:

$$\min_{\mathbf{Z}^i} F(\mathbf{Z}^i) = \frac{1}{2}\|\mathbf{x}_i - \mathbf{DZ}^i\|_2^2 + \lambda\|\mathbf{Z}^i\|_1 + \gamma\mathbf{R_A}(\mathbf{Z}^i) \tag{8}$$

where $\mathbf{R_A}(\mathbf{Z}^i) = \sum\limits_{j=1}^{n} \mathbf{A}_{ij}d(\mathbf{Z}^i, \mathbf{Z}^j)$.

Inspired by recent advances in solving non-convex optimization problems by proximal linearized method (Bolte et al., 2014), proximal gradient descent method (PGD) is used to optimize the nonconvex problem (8). Although the proximal mapping is typically associated with a lower semicontinuous function (Bolte et al., 2014) and it can be verified that $\mathbf{R_A}$ is not always lower semicontinuous, we can still derive a PGD-styple iterative method to optimize (8).

Define $\mathbf{G^A} \in \mathbb{R}^{p \times n}$ as $\mathbf{G}_{ki}^{\mathbf{A}} = \sum\limits_{j=1}^{n} \mathbf{A}_{ij}\mathbb{I}_{\mathbf{Z}_{kj}=0} - \sum\limits_{j=1}^{n} \mathbf{A}_{ij}\mathbb{I}_{\mathbf{Z}_{kj}\neq 0}$ where $\mathbb{I}$ is the indicator function, then $\mathbf{G}_{ki}^{\mathbf{A}}$ indicates the degree to which $\mathbf{Z}_{ki}$ is discouraged to be nonzero and it can be verified that [1]

$$\mathbf{R_A}(\mathbf{Z}^i) = \sum_{k=1}^{p} \mathbf{G}_{ki}^{\mathbf{A}}\mathbb{I}_{\mathbf{Z}_{ki}\neq 0} \tag{9}$$

Since each indicator function $\mathbb{I}_{\mathbf{Z}_{ki}\neq 0}$ is lower semicontinuous, $\mathbf{R_A}$ is lower semicontinuous if $\mathbf{G}_{ki}^{\mathbf{A}} \geq 0$ for $k = 1, \ldots, p$. In the following text, we let $Q(\mathbf{Z}^i) = \frac{1}{2}\|\mathbf{x}_i - \mathbf{DZ}^i\|_2^2$. The superscript with bracket indicates the iteration number of PGD or the iteration number of the coordinate descent without confusion. The PGD-style iterative method for optimizing (8) is as follows:

$$\tilde{\mathbf{Z}}^{i^{(t)}} = \mathbf{Z}^{i^{(t-1)}} - \frac{1}{\tau s}(\mathbf{D}^\top\mathbf{DZ}^{i^{(t-1)}} - \mathbf{D}^\top\mathbf{x}_i) \tag{10}$$

$$\mathbf{Z}_{ki}^{(t)} = \begin{cases} \arg\min\limits_{v \in \{\mathbf{u}_k, 0\}} H_k(v) & : \quad \mathbf{u}_k \neq 0 \text{ or } \mathbf{u}_k = 0 \text{ and } \mathbf{G}_{ki}^{\mathbf{A}} \geq 0 \\ \varepsilon & : \quad \mathbf{u}_k = 0 \text{ and } \mathbf{G}_{ki}^{\mathbf{A}} < 0 \end{cases} \tag{11}$$

for $k = 1, \ldots, p$ and $\varepsilon$ is any real number such that $\varepsilon \neq 0$ and $H_k(\varepsilon) \leq H_k(\mathbf{Z}_{ki}^{(t-1)})$. $H_k$ and $\mathbf{u}$ are defined below:

$$H_k(v) = \frac{\tau s}{2}(v - \tilde{\mathbf{Z}}_{ki}^{(t)})^2 + \lambda|v| + \gamma\mathbf{G}_{ki}^{\mathbf{A}}\mathbb{I}_{v\neq 0} \tag{12}$$

for $v \in \mathbb{R}$ and each $1 \leq k \leq p$, and

$$\mathbf{u} = \max\{|\tilde{\mathbf{Z}}^{i^{(t)}}| - \frac{\lambda}{\tau s}, 0\} \circ \text{sign}(\tilde{\mathbf{Z}}^{i^{(t)}}) \tag{13}$$

where $\circ$ means element-wise multiplication.

Proposition 1 shows that the PGD-style iterative method decreases the value of the objective function in each iteration.

---

[1] $\mathbf{R_A}(\mathbf{Z}^i)$ is equal to the right hand side of (9) up to a constant.

**Proposition 1.** *Let the sequence $\{\mathbf{Z}^{i(t)}\}_t$ be generated by the PGD-style iterative method with (10) and (11), then the sequence of the objective $\{F(\mathbf{Z}^{i(t)})\}_t$ decreases, and the following inequality holds for $t \geq 1$:*

$$F(\mathbf{Z}^{i(t)}) \leq F(\mathbf{Z}^{i(t-1)}) - \frac{(\tau-1)s}{2}\|\mathbf{Z}^{i(t)} - \mathbf{Z}^{i(t-1)}\|_2^2 \tag{14}$$

*And it follows that the sequence $\{F(\mathbf{Z}^{i(t)})\}_t$ converges.*

**Remark 1.** *(10) and (11) in each iteration of the proposed PGD-style iterative method resemble that of the ordinary PGD. (10) performs gradient descent on the differential part, and (11) can be viewed as an approximate solution to the proximal mapping $\min_{\mathbf{v}\in\mathbb{R}^p} H(\mathbf{v}) = \frac{\tau s}{2}\|\mathbf{v} - \tilde{\mathbf{Z}}^{i(t)}\|_2^2 + \lambda\|\mathbf{v}\|_1 + \gamma\mathbf{R_A}(\mathbf{v})$. Since $\mathbf{R_A}(\mathbf{Z}^i)$ is not always lower semicontinuous, $\arg\min_{\mathbf{v}\in\mathbb{R}^p} H(\mathbf{v})$ is not guaranteed to exist. One can see a simple example where this happens when $\mathbf{u}_k = 0$ and $\mathbf{G}_{ki}^{\mathbf{A}} < 0$ for some $k = 1,\ldots,p$, and in this case $\inf_{v\in\mathbb{R}} H_k(v) = \frac{\tau s}{2}(\tilde{\mathbf{Z}}_{ki}^{(t)})^2 + \gamma\mathbf{G}_{ki}^{\mathbf{A}}$ but this infinitum can not be achieved.*

In (10), $\tau > 1$ is a constant and $s$ is the Lipschitz constant for the gradient of function $Q(\cdot)$, namely

$$\|\nabla Q(\mathbf{y}) - \nabla Q(\mathbf{z})\|_2 \leq s\|\mathbf{y} - \mathbf{z}\|_2, \ \forall \mathbf{y}, \mathbf{z} \in \mathbb{R}^p \tag{15}$$

The PGD-style iterative method starts from $t = 1$ and continues until the sequence $\{F(\mathbf{Z}^{i(t)})\}$ converges or maximum iteration number is achieved. When the iterative method converges or terminates for each $\mathbf{Z}^i$, the step of coordinate descent for $\mathbf{Z}^i$ is finished and the optimization algorithm proceeds to optimize other sparse codes.

---

**Algorithm 1** Support Regularized Sparse Coding

**Input:**
> The data set $\boldsymbol{X} = \{\mathbf{x}_i\}_{i=1}^n$, the parameter $\lambda$, $\gamma$, maximum iteration number $M$ for the alternating method over $\mathbf{D}$ and $\mathbf{Z}$, and maximum iteration number $M_z$ for coordinate descent on $\mathbf{Z}$, maximum iteration number $M_p$ for the PGD-style iterative method on each $\mathbf{Z}^i$ (i = 1,...,n).
> and stopping threshold $\varepsilon$.

1: $m = 1$
2: **while** $m \leq M$ **do**
3: Perform coordinate descent to optimize (7) and obtain $\mathbf{Z}^{(m)}$ with fixed $\mathbf{D}^{(m-1)}$. In $i$-th ($1 \leq i \leq n$) step of each iteration of coordinate descent, solve (8) using the PGD-style iterative method (10) and (11) to update $\mathbf{Z}^i$ in each iteration of the PGD-style iterative method.
4: Optimize (6) using Lagrangian dual and obtain $\mathbf{D}^{(m)}$ with fixed $\mathbf{Z}^{(m)}$.
5: **if** $|L(\mathbf{D}^{(m)}, \mathbf{Z}^{(m)}) - L(\mathbf{D}^{(m-1)}, \mathbf{Z}^{(m-1)})| < \varepsilon$ **then**
6: **break**
7: **else**
8: $m = m + 1$.
9: **end if**
10: **end while**
**Output:** the support regularized sparse codes $\hat{\mathbf{Z}}$ when the above iterations converge or maximum iteration number is achieved.

---

### TIME COMPLEXITY

Algorithm 1 describes the algorithm of SRSC. We solve the ordinary sparse coding problem (1) by the online dictionary learning method (Mairal et al., 2009) and use the dictionary and the sparse codes as the initialization $\mathbf{D}^{(0)}$ and $\mathbf{Z}^{(0)}$ for the alternating method in Algorithm 1. In Algorithm 1, the time complexity of optimization over the sparse codes is $\mathcal{O}(MM_zM_pndp^2)$, and time complexity of optimization over the dictionary using Newton's method to solve the Lagrangian dual problem is $\mathcal{O}\Big(M\big(np^2 + T_{\text{newton}}(3p^{2.807} + 2dp^2 + dnp)\big)\Big)$, where $T_{\text{newton}}$ is the maximum iteration number for Newton's method. Therefore, the overall time complexity of Algorithm 1 is $\mathcal{O}\Big(M\big(M_zM_pndp^2 + np^2 + T_{\text{newton}}(3p^{2.807} + 2dp^2 + dnp)\big)\Big)$. It should be emphasized that the optimization over the dictionary for SRSC has the same efficiency as the efficient sparse coding method (Lee et al., 2006),

and the optimization over the sparse code of each data point by the PGD-style iterative method (10) and (11) is almost as efficient as the widely used Iterative Shrinkage and Thresholding Algorithm (ISTA) (Daubechies et al., 2004; Beck & Teboulle, 2009). Note that step (10) and (13) are required by both our method and ISTA; compared to ISTA, the extra operations incurred by our PGD-style iterative method (10) and (11) are only the arithmetic operations with time complexity $20p$ for evaluating the function $H_k(\cdot)$ defined in (12) for $k = 1, \ldots, p$. More specifically, evaluating the value of function $H_k(v)$ takes 10 arithmetic operations and two evaluations at $v = \mathbf{u}_k$ and $v = 0$ are needed. Since a compact dictionary is preferred by the extensive study of the sparse coding and dictionary learning literature and the dictionary size $p \leq 500$ is adopted throughout our experiments, our PGD-style iterative method only incurs extra operations of constant time complexity compared to ISTA while learning supported regularized sparse codes. In Section 4, we propose Deep-SRSC as a fast approximation of SRSC with considerable speedup for obtaining the sparse codes of the new data or the test data (see more details in Section 4.1). Furthermore, we conduct the empirical study and show that the parallel coordinate descent method, which updates the codes of a group of $P$ data points in parallel and provides $P$ times speedup over the coordinate descent method used in Section 2.1.2 and Algorithm 1, exhibits almost the same performance as the coordinate descent method for the clustering task on the test set of the CIFAR-10 data. Please refer to the details in the subsection "Deep-SRSC with the Second Test Setting (Referring to the Training Data)" in the Appendix.

## 2.2 RELATED WORK: $\ell^2$ REGULARIZED SPARSE CODING ($\ell^2$-RSC)

The manifold assumption (Belkin et al., 2006) is usually employed by existing regularized sparse coding methods (Liu et al., 2010; He et al., 2011; Zheng et al., 2011; Gao et al., 2013) to obtain the sparse code according to the manifold structure of the data. Interpreting the sparse code of a data point as its embedding, the manifold assumption in the case of sparse coding for most existing methods requires that if two points $\mathbf{x}_i$ and $\mathbf{x}_j$ are close in the intrinsic geometry of the submanifold, their corresponding sparse codes $\mathbf{Z}^i$ and $\mathbf{Z}^j$ are also expected to be similar to each other in the sense of $\ell^2$-distance (Zheng et al., 2011; Gao et al., 2013). In other words, $\mathbf{z}$ varies smoothly along the geodesics in the intrinsic geometry. Based on the spectral graph theory (Chung, 1997), extensive literature uses graph Laplacian to impose local smoothness of the embedding and preserve the local manifold structure (Belkin et al., 2006; Zheng et al., 2011; Gao et al., 2013).

The sparse code $\mathbf{Z}$ that captures the local geometric structure of the data in accordance with the manifold assumption by graph Laplacian minimizes the following $\ell^2$ regularization term, or the Laplacian regularization term:

$$\mathbf{R}_{\mathbf{A}}^{(\ell^2)}(\mathbf{Z}) = \frac{1}{2} \sum_{i=1}^{n} \sum_{j=1}^{n} \mathbf{A}_{ij} \|\mathbf{Z}^i - \mathbf{Z}^j\|_2^2 \tag{16}$$

where the $\ell^2$-norm is used to measure the distance between sparse codes, and $\mathbf{A}$ is the same as that in Section 2.1. $\mathbf{L_A} = \mathbf{D_A} - \mathbf{A}$ is the graph Laplacian associated with the similarity matrix $\mathbf{A}$, the degree matrix $\mathbf{D_A}$ is a diagonal matrix with each diagonal element being the sum of the elements in the corresponding row of $\mathbf{A}$, namely $(\mathbf{D_A})_{ii} = \sum_{j=1}^{n} \mathbf{A}_{ij}$. To the best of our knowledge, such $\ell^2$ regularization is employed by most methods that use graph regularization for sparse coding. Incorporating the $\ell^2$ regularization term into the optimization problem of sparse coding (1), the formulation of $\ell^2$ Regularized Sparse Coding ($\ell^2$-RSC) is presented below

$$\min_{\mathbf{D}, \mathbf{Z}} L^{(\ell^2)}(\mathbf{Z}) = \sum_{i=1}^{n} \|\mathbf{x}_i - \mathbf{D}\mathbf{Z}^i\|_2^2 + \lambda \|\mathbf{Z}^i\|_1 + \gamma^{(\ell^2)} \mathbf{R}_{\mathbf{A}}^{(\ell^2)}(\mathbf{Z}) \quad s.t. \ \|\mathbf{D}^k\|_2 \leq 1, k = 1, \ldots, p \tag{17}$$

### ADVANTAGE OF SRSC OVER $\ell^2$-RSC

Although $\ell^2$-RSC imposes the local smoothness on the sparse codes, it does not capture the locally linear structure of the data manifold. By promoting the smoothness on the support of the sparse codes rather than their $\ell^2$-distance, SRSC encodes the locally linear structure of the manifold in the sparse codes while reserving freedom in the sparse representation of the data with no constraints on the

magnitude of the sparse codes. Moreover, as pointed out by (Wang et al., 2015), support regularization offers robustness to noise for sparse coding. In SRSC, all the data consult their neighbors for choosing the dictionary atoms rather than choosing the atoms on their own, and the sparse codes of the noisy data are suppressed since they are forced to choose similar or the same atoms as the nearby clean data instead of choosing the atoms in the interests of representing themselves.

# 3 THEORETICAL ANALYSIS

It can be observed that optimization by coordinate descent over the sparse code in Section 2.1.2 is important for the overall optimization of SRSC, and each step of the coordinate descent (8) is a difficult nonconvex problem and crucial for obtaining the support regularized sparse code, where the nonconvexity comes from the support regularization term $\mathbf{R_A}(\mathbf{Z}^i)$ (9). Therefore, the optimization of (8) plays an important role in the overall optimization of SRSC. In the previous section, a PGD-style iterative method is proposed to decrease the value of the objective in each iteration. In this section, we provide further theoretical analysis on the optimization of problem (8) when $\mathbf{G}_{ki}^{\mathbf{A}} \geq 0$ for $k = 1, \ldots, p$. This condition is equivalent to the condition that the support regularization function

$$\mathbf{R_c}(\mathbf{v}) \triangleq \sum_{k=1}^{p} \mathbf{c}_k \mathbb{I}_{\mathbf{v}_k \neq 0} \qquad (18)$$

is lower semicontinuous, where $\mathbf{c} \in \mathbb{R}^p$ is the coefficients and $\mathbf{c}_k = \mathbf{G}_{ki}^{\mathbf{A}}$. Under this condition, we prove that the sequence $\{\mathbf{Z}^{i(t)}\}_t$ produced by the PGD-style iterative method converges to the sub-optimal solution which is a critical point of the objective (8). By connecting the support regularized function to the capped-$\ell^1$ norm and the nonconvexity analysis of the support regularization term, we present the bound for $\ell^2$-distance between the sub-optimal solution and the globally optimal solution to (8) in Theorem 1. Note that our analysis is valid for all $1 \leq i \leq n$.

We first have the following result that the support regularization function (18) is lower semicontinuous if and only if all the coefficients $\mathbf{c}$ are nonnegative.

**Proposition 2.** *The support regularization function (18) is lower semicontinuous if and only if all the coefficients* $\mathbf{c}$ *are nonnegative.*

Therefore, if $\mathbf{G}_{ki}^{\mathbf{A}} \geq 0$ for $k = 1, \ldots, p$, the support regularization term $\mathbf{R_A}(\mathbf{Z}^i)$ is lower semicontinuous with respect to $\mathbf{Z}^i$ in (9). In this case, the PGD-style iterative method proposed in Section 2.1.2 for each iteration $t \geq 1$ becomes

$$\tilde{\mathbf{Z}}^{i(t)} = \mathbf{Z}^{i(t-1)} - \frac{1}{\tau s}(\mathbf{D}^\top \mathbf{D} \mathbf{Z}^{i(t-1)} - \mathbf{D}^\top \mathbf{x}_i) \qquad (19)$$

$$\mathbf{Z}_{ki}^{(t)} = \underset{v \in \{\mathbf{u}_k, 0\}}{\arg \min} H_k(v), k = 1, \ldots, p \qquad (20)$$

which is equivalent to the updates rules in the ordinary proximal gradient descent method. It is worthwhile to mention the meaning of the condition that $\mathbf{G}_{ki}^{\mathbf{A}} \geq 0$ for $k = 1, \ldots, p$. For a data point $\mathbf{x}_i$, if the number of its neighbors with zero $k$-th element of the sparse codes is larger than that with nonzero $k$-th element of the sparse codes, which indicates that the neighbors of $\mathbf{x}_i$ suggest that a zero $k$-th element of the sparse code of $\mathbf{x}_i$ is preferable, then $\mathbf{G}_{ki}^{\mathbf{A}} \geq 0$ and $\mathbf{G}_{ki}^{\mathbf{A}}$ quantitatively represents the penalty if the sparse code element $\mathbf{Z}_k^i$ is nonzero while the neighbors of $\mathbf{x}_i$ suggest that $\mathbf{Z}_k^i = 0$ is preferable. Intuitively, this situation happens when there is conflict between choosing the support of the code solely by the data point itself and the suggestion of its neighbors; if the point is an outlier or suffering from noise, the optimization can help that point make a sensible choice by considering the suggestion of its neighbors. We observe that $\mathbf{G}_{ki}^{\mathbf{A}} \geq 0$ for $k = 1, \ldots, p$ happens in all the data sets used in this paper.

In the following lemma, we show that the sequence $\{\mathbf{Z}^{i(t)}\}_t$ generated by (19) and (20) converges to a critical point of $F(\mathbf{Z}^i)$, denoted by $\hat{\mathbf{Z}}^i$. Denote by $\mathbf{Z}^{i^*}$ the globally optimal solution to the original optimization problem (8). The following lemma also shows that both $\hat{\mathbf{Z}}^i$ and $\mathbf{Z}^{i^*}$ are local solutions to the capped-$\ell^1$ regularized problem (21). Before stating the lemma, the following definitions are introduced which are essential for our analysis.

**Definition 1.** *(Critical points) Given the non-convex function* $f \colon \mathbb{R}^n \to R \cup \{+\infty\}$ *which is a proper and lower semi-continuous function.*

- *for a given* $\mathbf{x} \in \mathrm{dom} f$, *its Frechet subdifferential of* $f$ *at* $\mathbf{x}$, *denoted by* $\tilde{\partial} f(x)$, *is the set of all vectors* $\mathbf{u} \in \mathbb{R}^n$ *which satisfy*

$$\limsup_{\mathbf{y} \neq \mathbf{x}, \mathbf{y} \to \mathbf{x}} \frac{f(\mathbf{y}) - f(\mathbf{x}) - \langle \mathbf{u}, \mathbf{y} - \mathbf{x} \rangle}{\|\mathbf{y} - \mathbf{x}\|} \geq 0$$

- *The limiting-subdifferential of* $f$ *at* $\mathbf{x} \in \mathbb{R}^n$, *denoted by written* $\partial f(x)$, *is defined by*

$$\partial f(x) = \{\mathbf{u} \in \mathbb{R}^n \colon \exists \mathbf{x}^k \to \mathbf{x}, f(\mathbf{x}^k) \to f(\mathbf{x}), \tilde{\mathbf{u}}^k \in \tilde{\partial} f(\mathbf{x}_k) \to \mathbf{u}\}$$

*The point* $\mathbf{x}$ *is a critical point of* $f$ *if* $0 \in \partial f(x)$.

Also, we are considering the following capped-$\ell^1$ regularized problem, which replaces the indicator function in the support regularization term $\mathbf{R}_{\mathbf{A}}(\mathbf{Z}^i)$ with the continuous capped-$\ell^1$ regularization term $\mathbf{T}$:

$$\min_{\boldsymbol{\beta} \in \mathbb{R}^p} L_{\text{capped}-\ell^1}(\boldsymbol{\beta}) = \frac{1}{2}\|\mathbf{x}_i - \mathbf{D}\boldsymbol{\beta}\|_2^2 + \lambda\|\boldsymbol{\beta}\|_1 + \mathbf{T}(\boldsymbol{\beta}; b) \tag{21}$$

where $\mathbf{T}(\boldsymbol{\beta}; b) = \sum\limits_{k=1}^{p} T_k(\boldsymbol{\beta}_k; b)$, $T_k(t; b) = \gamma \mathbf{G}_{ki}^{\mathbf{A}} \frac{\min\{|t|, b\}}{b}$ for some $b > 0$. It can be seen that the objective function of the capped-$\ell^1$ problem approaches that of (8) when $\frac{\min\{|t|, b\}}{b}$ approaches the indicator function $\mathbb{1}_{t \neq 0}$ as $b \to 0+$. Define $\mathbf{P}(\cdot; b) = \lambda\|\cdot\|_1 + \mathbf{T}(\cdot; b)$, the location solution to the capped-$\ell^1$ problem is defined as follows.

**Definition 2.** *(Local solution) A vector* $\tilde{\boldsymbol{\beta}}$ *is a local solution to the problem (21) if*

$$\|\mathbf{D}^{\top}(\mathbf{D}\tilde{\boldsymbol{\beta}} - \mathbf{x}_i) + \dot{\mathbf{P}}(\tilde{\boldsymbol{\beta}}; b)\|_2 = 0 \tag{22}$$

*where* $\dot{\mathbf{P}}(\tilde{\boldsymbol{\beta}}; b) = [\dot{P}_1(\tilde{\boldsymbol{\beta}}_1; b), \dot{P}_2(\tilde{\boldsymbol{\beta}}_2; b), \ldots, \dot{P}_p(\tilde{\boldsymbol{\beta}}_p; b)]^{\top}$, $P_k(t; b) = \lambda|t| + T_k(t; b)$ *for* $k = 1, \ldots, p$.

Note that in the above definition and the following text, $\dot{P}_k(t; b)$ can be chosen as any value between the right differential $\frac{\partial P_k}{\partial t}(t+; b)$ (or $\dot{P}_k(t+; b)$) and left differential $\frac{\partial P_k}{\partial t}(t-; b)$ (or $\dot{P}_k(t-; b)$) for $k = 1, \ldots, p$.

**Definition 3.** *(Degree of Nonconvexity of a Regularizer) For* $\kappa \geq 0$ *and* $t \in \mathbb{R}$, *define*

$$\theta(t, \kappa) := \sup_s \{-\mathrm{sgn}(s - t)(\dot{P}(s; b) - \dot{P}(t; b)) - \kappa|s - t|\}$$

*as the degree of nonconvexity for function* $P$. *If* $\mathbf{u} = (u_1, \ldots, u_p)^{\top} \in \mathbb{R}^p$, $\theta(\mathbf{u}, \kappa) = [\theta(u_1, \kappa), \ldots, \theta(u_p, \kappa)]$. *sgn is the sign function.*

Note that $\theta(t, \kappa) = 0$ for convex function $P$.

Let $\hat{\mathbf{S}}_i = \mathrm{supp}(\hat{\mathbf{Z}}^i)$ where $\mathrm{supp}(\cdot)$ indicates the support of a vector, i.e. the indices of its nonzero elements. Denote by $\mathbf{Z}^{i*}$ the globally optimal solution to (8), and $\mathbf{S}_i^* = \mathrm{supp}(\mathbf{Z}^{i*})$, then we have

**Lemma 1.** *For any* $1 \leq i \leq n$, *if* $\mathbf{G}_{ki}^{\mathbf{A}} \geq 0$ *for* $k = 1, \ldots, p$, *then the sequence* $\{\mathbf{Z}^{i(t)}\}_t$ *generated by (10) and (11) converges to a critical point of* $F(\mathbf{Z}^i)$, *which is denoted by* $\hat{\mathbf{Z}}^i$. *Moreover, if*

$$0 < b < \min\{\min_{j \in \hat{\mathbf{S}}_i} |\hat{\mathbf{Z}}_j^i|, \max_{k \notin \hat{\mathbf{S}}_i, \mathbf{G}_{ki}^{\mathbf{A}} \neq 0} \frac{\gamma \mathbf{G}_{ki}^{\mathbf{A}}}{(\frac{\partial Q}{\partial \mathbf{Z}_k^i}|_{\mathbf{Z}^i = \hat{\mathbf{Z}}^i} - \lambda)_+}, \min_{j \in \mathbf{S}_i^*} |\mathbf{Z}_j^{i*}|, \max_{k \notin \mathbf{S}_i^*, \mathbf{G}_{ki}^{\mathbf{A}} \neq 0} \frac{\gamma \mathbf{G}_{ki}^{\mathbf{A}}}{(\frac{\partial Q}{\partial \mathbf{Z}_j^i}|_{\mathbf{Z}^i = \mathbf{Z}^{i*}} - \lambda)_+}\} \tag{23}$$

*(if the denominator is* $0$, $\frac{\cdot}{0}$ *is defined to be* $+\infty$ *in the above inequality), then both* $\hat{\mathbf{Z}}^i$ *and* $\mathbf{Z}^{i*}$ *are local solutions to the capped-$\ell^1$ regularized problem (21).*

Using the degree of nonconvexity of the regularizer $\mathbf{P}$, we have the following theorem showing that the sub-optimal solution $\hat{\mathbf{Z}}^i$ obtained by our PGD-style iterative method can be close to the globally optimal solution to the original problem (8), i.e. $\mathbf{Z}^{i^*}$. In the following text, $\mathbf{B_I}$ indicates a submatrix of $\mathbf{B}$ whose columns correspond to the nonzero elements of $\mathbf{I}$, and $\sigma_{\min}(\cdot)$ indicates the smallest singular value of a matrix.

**Theorem 1.** *(Sub-optimal solution is close to the globally optimal solution) For any $1 \le i \le n$, let $\mathbf{E}_i = \hat{\mathbf{S}}_i \cup \mathbf{S}_i^*$. Suppose $\mathbf{G}_{ki}^{\mathbf{A}} \ge 0$ for $k = 1, \ldots, p$, $\mathbf{D}_{\mathbf{E}_i}$ is not singular with $\kappa_0 \triangleq \sigma_{\min}(\mathbf{D}_{\mathbf{E}_i}) > 0$, $\kappa_0^2 > \kappa > 0$, and $b$ is chosen according to (23) as in Lemma 1. Let $\tilde{\mathbf{S}}_i = (\hat{\mathbf{S}}_i \setminus \mathbf{S}_i^*) \cup (\mathbf{S}_i^* \setminus \hat{\mathbf{S}}_i)$ be the symmetric difference between $\hat{\mathbf{S}}_i$ and $\mathbf{S}_i^*$, then*

$$\|\mathbf{Z}^{i^*} - \hat{\mathbf{Z}}^i\|_2 \le \frac{1}{\kappa_0^2 - \kappa}\left( \Big( \sum_{k \in \tilde{\mathbf{S}}_i \cap \hat{\mathbf{S}}_i} (\max\{0, \frac{\gamma \mathbf{G}_{ki}^{\mathbf{A}}}{b} - \kappa|\hat{\mathbf{Z}}_{ki} - b|\})^2 + \sum_{k \in \tilde{\mathbf{S}}_i \setminus \hat{\mathbf{S}}_i} (\max\{0, \frac{\gamma \mathbf{G}_{ki}^{\mathbf{A}}}{b} - \kappa b\})^2 \Big)^{\frac{1}{2}} + \|\mathbf{t}\|_2 \right)$$

(24)

*where $\mathbf{t} \in \mathbb{R}^p$, $\mathbf{t}_k = 2\lambda \mathbb{I}_{\mathbf{Z}_k^{i^*} * \hat{\mathbf{z}}_k^i < 0} + 0 \mathbb{I}_{\mathbf{Z}_k^{i^*} * \hat{\mathbf{z}}_k^i > 0}$ for $k \in \hat{\mathbf{S}}_i \cap \mathbf{S}_i^*$, and $\mathbf{t}_k = 0$ for all other $k$.*

**Remark 2.** *Note that the bound for distance between the sub-optimal solution and the globally optimal solution presented in Theorem 1 does not require typical Restricted Isometry Property (RIP) conditions, e.g. Cands (2008). Also, when $\frac{\gamma \mathbf{G}_{ki}^{\mathbf{A}}}{b} - \kappa|\hat{\mathbf{Z}}_{ki} - b|$ and $\frac{\gamma \mathbf{G}_{ki}^{\mathbf{A}}}{b} - \kappa b$ are no greater than $0$ and $\mathbf{Z}^{i^*}$ and $\hat{\mathbf{Z}}^i$ has the same sign in the intersection of their support, the sub-optimal solution $\hat{\mathbf{Z}}^i$ is equal to the globally optimal solution. When $\frac{\gamma \mathbf{G}_{ki}^{\mathbf{A}}}{b} - \kappa|\hat{\mathbf{Z}}_{ki} - b|$ and $\frac{\gamma \mathbf{G}_{ki}^{\mathbf{A}}}{b} - \kappa b$ are small positive numbers and $\mathbf{Z}^{i^*}$ and $\hat{\mathbf{Z}}^i$ has similar sign in the intersection of their support, $\hat{\mathbf{Z}}^i$ is close to the globally optimal solution.*

## 4 DEEP SUPPORT REGULARIZED SPARSE CODING: DEEP-SRSC

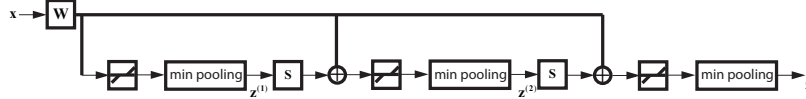

Figure 3: Illustration of Deep-SRSC for approximate Support Regularized Sparse Coding.

Inspired by the PGD-style iterative method (10) and (11) for SRSC and the LISTA network, we propose Deep Support Regularized Sparse Coding (Deep-SRSC) illustrated in Figure 3, which is a neural network that produces the approximate support regularized sparse codes for SRSC. The goal of Deep-SRSC is to approximate the sparse codes of the input data in a fast way by feeding the data through the Deep-SRSC network, instead of running the iterative optimization algorithm for SRSC in Section 2.1. To achieve this goal, the Deep-SRSC network is trained on the training data by minimizing the squared distance between the predicted codes of the training data by the network and their ground truth codes. The network design of Deep-SRSC is in accordance with the proposed PGD-style iterative method. When $\mathbf{W} = \frac{1}{L}\mathbf{D}^\top$, $\mathbf{S} = \mathbf{I} - \frac{1}{L}\mathbf{D}^\top\mathbf{D}$ where $L = \tau s$, then each stage in the recurrent structure of Deep-SRSC implements one iteration of PGD-style iterative method, i.e. (10) and (11). In Deep-SRSC, $\mathbf{W}$, $\mathbf{S}$ and $L$ are to be learned by the network rather than computed from a pre-computed dictionary $\mathbf{D}$, and $\mathbf{S}$ is shared over different layers. The min-pooling neuron in Deep-SRSC outputs the result of $\arg\min_{v \in \{\mathbf{u}_k, 0\}} H_k(v)$ or $\varepsilon$, according to the update rule (11). Figure 3 illustrates Deep-SRSC with 2 layers.

Denote the training data by $\mathbf{x}_1, \ldots, \mathbf{x}_m$, and let $\mathbf{Z}_{\mathrm{sr}}$ be the ground truth support regularized sparse codes of the training data which are obtained by the optimization method introduced in Section 2.1. Let $f_{\mathrm{sr}}$ be the Deep-SRSC encoder which produces the approximate support regularized sparse code $\mathbf{z} = f_{\mathrm{sr}}(\mathbf{x}, \Theta_{\mathrm{sr}})$, where $\Theta_{\mathrm{sr}} = (\mathbf{W}, \mathbf{S}, L)$ denotes the parameters of Deep-SRSC. Then the parameters of Deep-SRSC are learned by minimizing the cost function which measures the distance between the predicted approximate support regularized sparse codes and the ground truth ones: $\frac{1}{m}\sum_{i=1}^{m} \|\mathbf{Z}_{\mathrm{sr}}^i - f_{sr}(\mathbf{x}_i, \Theta_{\mathrm{sr}})\|_2^2$. Similar to the LISTA network, the optimization is performed by

stochastic gradient descent and back-propagation. The batch size is set to 1 so as to simulate the coordinate descent method for optimization over the sparse codes in Section 2.1.2. The adjacency matrix of the KNN graph over the training data is required as input for training the network.

The approximate codes of the new data, or the test data, are obtained by feeding the new data through the Deep-SRSC network learned on the training data. We provide two test settings below, depending on whether training data are referred to in the test process.

1) In the first setting where the training data are not referred to, the test data are a group of data points. The test data and the KNN graph over them are fed into the Deep-SRSC network to obtain the approximate codes of the test data. The locally linear manifold structure of the test data is encoded in the KNN graph over the test data. This setting is potentially more suitable for the situation of limited storage where the training data and their codes do not need to be stored in the test process. This setting may not be suitable for the test data that do not reliably reflect the locally linear manifold structure (e.g. in the case of a very small amount of test data), and in this case the second setting below is a better choice.

2) In the second setting where training data are referred to, the approximate code of each data point is obtained by feeding that point and the KNN graph over that point and the training data into the Deep-SRSC network. The code of each test point is reliably obtained by referring to its nearest neighbors in the training data and this process is independent of the factor that whether the test data reflect the locally linear manifold structure.

## 4.1 DEEP-SRSC AS FAST ENCODER

It should be emphasized that Deep-SRSC is a fast encoder for SRSC when obtaining the codes of the new data (or test data). Each layer of Deep-SRSC resembles one iteration of the PGD-style iterative method (10) and (11), and the computational cost of feeding forward a data point through one layer is the same as that of executing one iteration of the PGD-style iterative method for that point. Therefore, the feed-forward process of obtaining the sparse codes of the new data using $\ell$-layer Deep-SRSC is around $\frac{M_p}{\ell}$ times faster than the PGD-style iterative method used in Algorithm 1, where $M_p$ is the maximum iteration number for the PGD-style iterative method. In the experimental results shown in the next section, Deep-SRSC with different number of layers are employed to produce the approximate support regularized sparse codes, and 6-layer Deep-SRSC achieves minimum prediction error. With $M_p = 50$ throughout our experiments, Deep-SRSC is around $\frac{50}{6} \approx 8.3$ times faster than the PGD-style iterative method. Our analysis in this subsection holds for both test settings.

Table 1: Clustering results on USPS handwritten digits database. $c$ in the left column is the cluster number, i.e. the first $c$ clusters of the entire data are used for clustering.

| USPS # Clusters | Measure | KM | SC | Sparse Coding | $\ell^2$-RSC | SRSC |
|---|---|---|---|---|---|---|
| c = 4 | AC | 0.9243 | 0.4514 | 0.9869 | 0.9869 | **0.9880** |
| | NMI | 0.7782 | 0.4160 | 0.9429 | 0.9429 | **0.9467** |
| c = 6 | AC | 0.7130 | 0.4325 | 0.7781 | 0.7781 | **0.9723** |
| | NMI | 0.6845 | 0.4865 | 0.8507 | 0.8507 | **0.9135** |
| c = 8 | AC | 0.7294 | 0.4227 | 0.8163 | 0.8163 | **0.9645** |
| | NMI | 0.6851 | 0.4811 | 0.8669 | 0.8669 | **0.9027** |
| c = 10 | AC | 0.6878 | 0.4041 | 0.8178 | 0.8287 | **0.8293** |
| | NMI | 0.6312 | 0.4765 | 0.8321 | 0.8398 | **0.8471** |

Table 2: Clustering results on various data sets

| Data Set | Measure | KM | SC | Sparse Coding | $\ell^2$-RSC | SRSC |
|---|---|---|---|---|---|---|
| COIL-20 | AC | 0.6274 | 0.3347 | 0.9903 | 0.9903 | **0.9944** |
| | NMI | 0.7533 | 0.5667 | 0.9879 | 0.9879 | **0.9933** |
| COIL-100 | AC | 0.5221 | 0.2372 | 0.6979 | 0.6979 | **0.7267** |
| | NMI | 0.7633 | 0.5410 | 0.8837 | 0.8837 | **0.8876** |
| UCI Gesture Phase Segmentation | AC | 0.3868 | 0.3375 | 0.4003 | 0.4023 | **0.4123** |
| | NMI | 0.1191 | **0.1300** | 0.1164 | 0.1164 | 0.1187 |

Table 3: Prediction error (average squared error between the predicted codes and the ground truth codes) of Deep-SRSC with different depth and different dictionary size on the test set of USPS data, using the first test setting

| Dictionary Size | 1-layer | 2-layer | 6-layer |
|---|---|---|---|
| $p = 100$ | 0.06 | 0.04 | 0.04 |
| $p = 300$ | 0.14 | 0.09 | 0.07 |
| $p = 500$ | 0.24 | 0.12 | 0.11 |

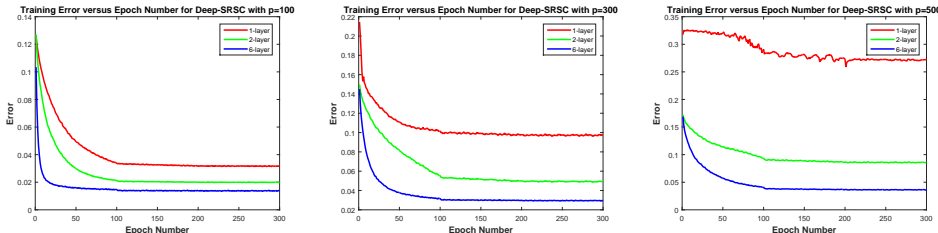

Figure 4: Training error of Deep-SRSC with dictionary size $p = 100$, $p = 300$, and $p = 500$. The test error of Deep-SRSC is shown in Figure 5 in the appendix.

Table 4: Clustering results on the test set of USPS data with different dictionary size $p$

| Dictionary Size | Measure | KM | SC | Sparse Coding | $\ell^2$-RSC | SRSC | 6-Layer Deep-SRSC |
|---|---|---|---|---|---|---|---|
| $p = 100$ | AC | 0.6020 | 0.3279 | 0.6363 | 0.6363 | 0.7105 | **0.7155** |
| | NMI | 0.5522 | 0.4372 | 0.7011 | 0.7011 | **0.7068** | 0.6778 |
| $p = 300$ | AC | - | - | 0.6408 | 0.6462 | **0.7225** | 0.7000 |
| | NMI | - | - | 0.7011 | 0.7011 | **0.7045** | 0.6817 |
| $p = 500$ | AC | - | - | 0.6263 | 0.6268 | 0.6248 | **0.6836** |
| | NMI | - | - | 0.6872 | 0.6898 | **0.7221** | 0.6537 |

# 5 EXPERIMENTAL RESULTS

## 5.1 CLUSTERING PERFORMANCE

In this subsection, the superiority of SRSC is demonstrated by its performance in data clustering on various data sets, e.g. USPS handwritten digits data set, COIL-20, COIL-100 and UCI Gesture Phase Segmentation data set. Two measures are used to evaluate the performance of the clustering methods, i.e. the Accuracy (AC) and the Normalized Mutual Information (NMI) (Zheng et al., 2004). SRSC is compared to K-means (KM), Spectral Clustering (SC), Sparse Coding and $\ell^2$-RSC in Section 2.2. Throughout all the experiments, we set $K = 3$ for building the adjacency matrix $\mathbf{A}$ of KNN graph, dictionary size $p = 300$ and $\lambda = 0.1$ for both $\ell^2$-RSC and SRSC. We also set $\gamma^{(\ell^2)} = 1$ which is the suggested default value in (Zheng et al., 2011), and $M = M_z = 5$ and $M_p = 50$ in Algorithm 1. The default value of the weight for support regularization term of SRSC is $\gamma = 0.5$. SRSC is implemented by both MATLAB and CUDA C++ with extreme efficiency, and the code is published on GitHub: https://github.com/yingzhenyang/SRSC.

The USPS handwritten digits data set is comprised of $n = 9298$ handwritten images of ten digits from 0 to 9, and each image is of size $16 \times 16$ and represented by a 256-dimensional vector. The whole data set is divided into training set of 7291 images and test set of 2007 images. We run Algorithm 1 to obtain the support regularized sparse code $\hat{\mathbf{Z}}$, then build a $n \times n$ similarity matrix $\mathbf{Y}$ over all the data. Two similarity measure are employed: the first similarity is the positive part of the inner product of their corresponding sparse codes, namely $\mathbf{Y}_{ij} = \max\{0, \hat{\mathbf{Z}}^{i\top}\hat{\mathbf{Z}}^j\}$, the second one is $\mathbf{Y}_{ij} = \mathbf{A}_{ij}\mathbf{q}_{\hat{\mathbf{Z}}^i}^{\top}\mathbf{q}_{\hat{\mathbf{Z}}^j}$ where $\mathbf{q}_{\mathbf{v}}$ is a binary vector of the same size as $\mathbf{v}$ with element 1 at the indices of nonzero elements of $\mathbf{v}$. The second similarity measure is name the support similarity and it considers the number of common dictionary atoms chosen by the sparse codes. Spectral clustering is performed on the similarity matrix $\mathbf{Y}$ to obtain the clustering result of SRSC, and the best performance among the two similarity measures is reported. The same procedure is performed by all the other sparse coding based methods to obtain clustering results. The clustering results of various methods are shown in Table 1.

COIL-20 Database has 1440 images of resolution $32 \times 32$ for 20 objects, and the background is removed in all images. The dimension of this data is 1024. Its enlarged version, COIL-100 Database, contains 100 objects with 72 images of resolution $32 \times 32$ for each object. The images of each object were taken 5 degrees apart when each object was rotated on a turntable. The UCI Gesture Phase Segmentation data set contains the gesture information of three users when they told stories of some comic strips in front of the Microsoft Kinect sensor. We use the processed file provided by the original data consisting of 9873 frames, and the gesture information in each frame is the vectorial velocity and acceleration of left hand, right hand, left wrist, and right wrist, represented by a 32-dimensional vector. The clustering results on these three data sets are shown in Table 2. It can be observed from Table 1 and Table 2 that SRSC always produces better clustering accuracy than other competing methods, due to its capability of capturing the locally linear manifold structure of the data and robustness to noies. In the appendix, we further show the performance of different sparse coding based methods with different dictionary size on COIL-100 data set in Table 5, and investigate the parameter sensitive of SRSC by demonstrating its performance with varying $\gamma$ and $K$ in Table 6.

## 5.2    Approximation by Deep-SRSC

In this subsection, Deep-SRSC is employed as a fast encoder to approximate the support regularized sparse codes of SRSC on the USPS data set. Throughout this subsection, we show results using the first test setting introduced in Section 4, i.e. test without referring to the training data. Additional experimental results on the performance of Deep-SRSC with the second test setting, including the application to semi-supervised learning by label propagation (Zhu et al., 2003), are shown in the appendix.

The Deep-SRSC network is trained on the training set of the USPS data comprising 7291 images. We adopt three depth settings wherein Deep-SRSC has 1 layer, 2 layers, and 6 layers respectively. We first run SRSC on the training set of USPS data to obtain the dictionary $\mathbf{D}_{\mathrm{sr}}$ and the support regularized sparse codes $\mathbf{Z}_{\mathrm{sr}}$ of the training data. Then the optimization problem (7) is solved by the PGD-style iterative method in Section 2.1.2, where $\boldsymbol{X}$ is the test data, $\mathbf{A}$ is the adjacency matrix of the KNN graph over the test data, to obtain the support regularized sparse codes $\mathbf{Z}_{\mathrm{sr,test}}$ of the test data with dictionary $\mathbf{D}_{\mathrm{sr}}$. $\mathbf{Z}_{\mathrm{sr}}$ is used as the ground truth support regularized sparse codes to train Deep-SRSC, and $\mathbf{Z}_{\mathrm{sr,test}}$ serves as the ground truth codes of the test data. The approximate codes of the test data of the USPS data are obtained by feeding forward them into the Deep-SRSC network together with the KNN graph over the test data, and the prediction error of Deep-SRSC is the average of the squared error between the predicted codes and $\mathbf{Z}_{\mathrm{sr,test}}$. Figure 4 illustrates the training error of Deep-SRSC w.r.t. the epoch number for 1 layer, 2 layers, and 6 layers respectively, and Figure 5 in the appendix illustrates the test error of Deep-SRSC. For each depth setting, Deep-SRSC is trained with 300 epoches, and testing is performed for every 5 epoches during training. It can be observed that deeper Deep-SRSC leads to smaller training and test error. Deep-SRSC is implemented with TensorFlow (Abadi et al., 2016). The initial learning rate is set to $10^{-4}$, and divided by 10 at 100-th epoch and 200-th epoch, so the final learning rate is $10^{-6}$ upon the termination of the training.

Table 3 shows the prediction error of Deep-SRSC for different dictionary size $p$ and different number of layers. It can be observed that Deep-SRSC with more layers demonstrates smaller prediction error for the same dictionary size due to its better representation capability, and smaller dictionary size leads to less prediction error for the same number of layers due to the reduced difficulty of representation. Moreover, the codes predicted by 6-layer Deep-SRSC are used to perform clustering on the test data because of its minimum prediction error, with comparison to the performance of sparse coding and $\ell^2$-RSC shown in Table 4 with respect to different dictionary size. For either sparse coding or $\ell^2$-RSC, the dictionary is firstly learned on the training data, then the sparse codes of the test data obtained with respect to that dictionary are used to perform clustering on the test set of USPS data. We can see that SRSC together with its approximation, 6-layer Deep-SRSC, achieve the highest accuracy and NMI. In addition, a reasonably large dictionary benefits SRSC, e.g. increasing $p$ from 100 to 300 boosts its accuracy, since the dictionary atoms serve as the basis for the locally linear structures (local subspaces) of the data manifold and a sufficiently large dictionary size is favorable for modeling all such locally linear structures. On the other hand, a too large dictionary (such as $p = 500$) imposes much difficulty on the optimization which can even hurt the performance of SRSC, $\ell^2$-RSC and regular sparse coding.

## 6 CONCLUSION

We propose Support Regularized Sparse Coding (SRSC) which captures the locally linear manifold structure of the high-dimensional data for sparse coding and enjoys robustness to noise. SRSC achieves this goal by encouraging nearby data in the manifold to share dictionary atoms. The optimization algorithm of SRSC is presented with theoretical guarantee for the optimization over the sparse codes. In addition, we propose Deep-SRSC, a feed-forward neural network, as a fast encoder to approximate the support regularized sparse codes produce by SRSC. Experimental results demonstrate the effectiveness of SRSC by its application to data clustering, and show that Deep-SRSC renders approximate codes for SRSC with low prediction error. The approximate codes generated by 6-layer Deep-SRSC also deliver compelling empirical performance for data clustering.

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

## APPENDIX

### PROOFS

***Proof of Proposition 1***. Note that $\mathbf{u}$ is the optimal solution to the lasso problem $\arg\min_{\mathbf{v} \in R^p} \frac{\tau s}{2} \|\mathbf{v} - \tilde{\mathbf{Z}}_{ki}^{(t)}\|_2^2 + \lambda\|\mathbf{v}\|_1$. Define $T_k(v) = \frac{\tau s}{2}(v - \tilde{\mathbf{Z}}_{ki}^{(t)})^2 + \lambda|v|$ for $v \in \mathbb{R}$, then $\mathbf{u}_k = \arg\min_{v \in \mathbb{R}} T_k(v)$. Since the

two functions $H_k(v)$ and $T_k(v)$ only differ at $v = 0$, $\arg\min_{v\in\{\mathbf{u}_k,0\}} H_k(v)$ is the optimal solution to $\min_{v\in R} H_k(v)$ when $\mathbf{u}_k \neq 0$ or $\mathbf{u}_k = 0$ and $\mathbf{G}^{\mathbf{A}}_{ki} \geq 0$.

When $\mathbf{u}_k = 0$ and $\mathbf{G}^{\mathbf{A}}_{ki} < 0$, when $\varepsilon \to 0$ and $\varepsilon \neq 0$, $H_k(v) \to \mathbf{G}^{\mathbf{A}}_{ki}$ and $\inf_{v\in\mathbb{R}} H_k(v) = \frac{\tau s}{2}(\tilde{\mathbf{Z}}^{(t)}_{ki})^2 + \gamma\mathbf{G}^{\mathbf{A}}_{ki}$. Note that the infimum can never be achieved. Since $\inf_{v\in\mathbb{R}} H_k(v) < H_k(\mathbf{Z}^{(t-1)}_{ki})$, we can always find $\varepsilon \neq 0$ such that $H_k(\varepsilon) \leq H_k(\mathbf{Z}^{(t-1)}_{ki})$.

Define $H(\mathbf{v}) = \frac{\tau s}{2}\|\mathbf{v} - \tilde{\mathbf{Z}}^{i^{(t)}}\|_2^2 + \lambda\|\mathbf{v}\|_1 + \gamma\mathbf{R}_{\mathbf{A}}(\mathbf{v})$. Based on the above argument, $H(\mathbf{Z}^{i^{(t)}}) \leq H(\mathbf{Z}^{i^{(t-1)}})$ which indicates that

$$\frac{\tau s}{2}\|\mathbf{Z}^{i^{(t)}} - \mathbf{Z}^{i^{(t-1)}}\|_2^2 + \langle \mathbf{Z}^{i^{(t)}} - \mathbf{Z}^{i^{(t-1)}}, \nabla Q(\mathbf{Z}^{i^{(t-1)}})\rangle \tag{25}$$

$$+ \lambda\|\mathbf{Z}^{i^{(t)}}\|_1 + \gamma\mathbf{R}_{\mathbf{A}}(\mathbf{Z}^{i^{(t)}}) \leq \lambda\|\mathbf{Z}^{i^{(t-1)}}\|_1 + \gamma\mathbf{R}_{\mathbf{A}}(\mathbf{Z}^{i^{(t-1)}}) \tag{26}$$

Also, since $s$ is the Lipschitz constant for the gradient of function $Q(\cdot)$, we have

$$Q(\mathbf{Z}^{i^{(t)}}) \leq Q(\mathbf{Z}^{i^{(t-1)}}) + \langle\mathbf{Z}^{i^{(t)}} - \mathbf{Z}^{i^{(t-1)}}, \nabla Q(\mathbf{Z}^{i^{(t-1)}})\rangle + \frac{s}{2}\|\mathbf{Z}^{i^{(t)}} - \mathbf{Z}^{i^{(t-1)}}\|_2^2 \tag{27}$$

Combining (25) and (27),

$$F(\mathbf{Z}^{i^{(t)}}) \leq F(\mathbf{Z}^{i^{(t-1)}}) - \frac{(\tau-1)s}{2}\|\mathbf{Z}^{i^{(t)}} - \mathbf{Z}^{i^{(t-1)}}\|_2^2$$

$\square$

***Proof of Lemma 1.*** We first prove that the sequences $\{\mathbf{Z}^{i^{(t)}}\}_t$ is bounded for any $1 \leq i \leq n$. By Proposition 1, the sequence $\{F(\mathbf{Z}^{i^{(t)}})\}_t$ decreases, so we have

$$F(\mathbf{Z}^{i^{(t)}}) = \frac{1}{2}\|\mathbf{x}_i - \mathbf{D}\mathbf{Z}^{i^{(t)}}\|_2^2 + \lambda\|\mathbf{Z}^{i^{(t)}}\|_1 + \gamma\mathbf{R}_{\mathbf{A}}(\mathbf{Z}^{i^{(t)}})$$
$$\leq \|\mathbf{x}_i - \mathbf{D}\mathbf{Z}^{i^{(0)}}\|_2^2 + \lambda\|\mathbf{Z}^{i^{(0)}}\|_0 \leq 1 + \mathbf{R}_{\mathbf{A}}(\mathbf{Z}^{i^{(0)}})$$

for $t \geq 1$. Therefore,

$$\|\mathbf{Z}^{i^{(t)}}\|_1 \leq \frac{1}{\lambda}\|\mathbf{x}_i - \mathbf{D}\mathbf{Z}^{i^{(0)}}\|_2^2 + \lambda\|\mathbf{Z}^{i^{(0)}}\|_0 \leq 1 + \mathbf{R}_{\mathbf{A}}(\mathbf{Z}^{i^{(0)}})$$

It follows that $\|\mathbf{Z}^{i^{(t)}}\|_1$ is bounded, and $\|\mathbf{Z}^{i^{(t)}}\|_2$ is also bounded. Since $\mathbf{G}^{\mathbf{A}}_{ki} \geq 0$ for $k = 1,\ldots,p$ and the indicator function $\mathbb{I}_{.\neq 0}$ is semi-algebraic function, $\mathbf{R}_{\mathbf{A}}(\cdot)$ is also a semi-algebraic function and lower semicontinuous. Therefore, according to Theorem 1 by Bolte et al. (2014), $\{\mathbf{Z}^{i^{(t)}}\}_t$ converges to a critical point of $F(\mathbf{Z}^i)$, denoted by $\hat{\mathbf{Z}}^i$.

Let $\hat{\mathbf{v}} = \mathbf{D}^\top(\mathbf{D}\hat{\mathbf{Z}}^i - \mathbf{x}_i) + \dot{\mathbf{P}}(\hat{\mathbf{Z}}^i; b)$. For $k$ such that $\mathbf{G}^{\mathbf{A}}_{ki} = 0$, since $\hat{\mathbf{Z}}^i$ is a critical point of $F(\mathbf{Z}^i)$, $\hat{\mathbf{v}}_k = 0$. Now we consider the case that $\mathbf{G}^{\mathbf{A}}_{ki} \neq 0$.

For for $k \in \hat{\mathbf{S}}_i$, since $\hat{\mathbf{Z}}^i$ is a critical point of $F(\mathbf{Z}^i) = \frac{1}{2}\|\mathbf{x}_i - \mathbf{D}\mathbf{Z}^i\|_2^2 + \lambda\|\mathbf{Z}^i\|_1 + \gamma\mathbf{R}_{\mathbf{A}}(\mathbf{Z}^i)$. then $\frac{\partial(Q+\lambda\|\mathbf{Z}^i\|_1)}{\partial\mathbf{Z}^i_k}|_{\mathbf{Z}^i=\hat{\mathbf{Z}}^i} = 0$ because $\frac{\partial\mathbf{R}_{\mathbf{A}}(\mathbf{Z}^i)}{\partial\mathbf{Z}^i_k}|_{\mathbf{Z}^i=\hat{\mathbf{Z}}^i} = 0$. Note that $\min_{k\in\hat{\mathbf{S}}_i}|\hat{\mathbf{Z}}^i_k| > b$, so $\frac{\partial\mathbf{T}}{\partial\mathbf{Z}^i_k}|_{\mathbf{Z}^i=\hat{\mathbf{Z}}^i} = 0$, and it follows that $\hat{\mathbf{v}}_k = 0$.

For $k \notin \hat{\mathbf{S}}_i$, since $\frac{dP_k}{d\mathbf{Z}^i_k}(\hat{\mathbf{Z}}^i_k+; b) = \frac{\gamma\mathbf{G}^{\mathbf{A}}_{ki}}{b} + \lambda$ and $\frac{dP_k}{d\mathbf{Z}^i_k}(\hat{\mathbf{Z}}^i_k-; b) = -\frac{\gamma\mathbf{G}^{\mathbf{A}}_{ki}}{b} - \lambda$, $\frac{\gamma\mathbf{G}^{\mathbf{A}}_{ki}}{b} + \lambda \geq |\frac{\partial Q}{\partial\mathbf{Z}^i_k}|_{\mathbf{Z}^i=\hat{\mathbf{Z}}^i}|$, we can choose the $k$-th element of $\dot{\mathbf{P}}(\hat{\mathbf{Z}}^i; b)$ such that $\hat{\mathbf{v}}_k = 0$. Therefore, $\|\hat{\mathbf{v}}\|_2 = 0$, and $\hat{\mathbf{Z}}^i$ is a local solution to the problem (21).

Now we prove that $\mathbf{Z}^{i^*}$ is also a local solution to (21). Let $\mathbf{v}^* = \mathbf{D}^\top(\mathbf{D}\mathbf{Z}^{i^*} - \mathbf{x}_i) + \dot{\mathbf{P}}(\mathbf{Z}^{i^*}; b)$, and $Q$ is defined as before. For $k$ such that $\mathbf{G}^{\mathbf{A}}_{ki} = 0$, since $\mathbf{Z}^{i^*}$ is the globally optimal solution of $F(\mathbf{Z}^i)$, $\mathbf{v}^*_k = 0$.

Again we consider the case that $\mathbf{G}^{\mathbf{A}}_{ki} \neq 0$.

For $k \in \mathbf{S}^*_i$, since $\mathbf{Z}^{i^*}$ is the globally optimal solution to problem (8), we also have $\frac{\partial(Q+\lambda\|\mathbf{Z}^i\|_1)}{\partial\mathbf{Z}^i_k}|_{\mathbf{Z}^i=\mathbf{Z}^{i^*}} = 0$. If it is not the case and $\frac{\partial(Q+\lambda\|\mathbf{Z}^i\|_1)}{\partial\mathbf{Z}^i_k}|_{\mathbf{Z}^i=\mathbf{Z}^{i^*}} \neq 0$, then we can change $\mathbf{Z}^i_k$ by a small amount in the direction of the gradient $\frac{\partial(Q+\lambda\|\mathbf{Z}^i\|_1)}{\partial\mathbf{Z}^i_k}$ at the point $\mathbf{Z}^i = \mathbf{Z}^{i^*}$ while $\mathbf{Z}^i_k$ is still nonzero, leading to a smaller value of the objective $F(\mathbf{Z}^i)$.

Note that $\min_{k \in \mathbf{S}_i^*} |\mathbf{Z}_k^{i\,*}| > b$, so $\frac{\partial \mathbf{T}}{\partial \mathbf{Z}_k^i}|_{\mathbf{Z}^i = \hat{\mathbf{z}}^i} = 0$, and it follows that $\mathbf{v}_k^* = 0$.

For $k \notin \mathbf{S}_i^*$, since $\frac{\gamma \mathbf{G}_{ki}^{\mathbf{A}}}{b} + \lambda \geq \max_{k \notin \hat{\mathbf{S}}_i} |\frac{\partial Q}{\partial \mathbf{Z}_k^i}|_{\mathbf{Z}^i = \mathbf{Z}^{i\,*}}|$, we can choose the $k$-th element of $\dot{\mathbf{P}}(\mathbf{Z}^{i\,*}; b)$ such that $\mathbf{v}_k^* = 0$. It follows that $\|\mathbf{v}^*\|_2 = 0$, and $\mathbf{Z}^{i\,*}$ is also a local solution to the problem (21).  □

***Proof of Theorem 1***. According to Lemma 1, both $\hat{\mathbf{Z}}^i$ and $\mathbf{Z}^{i\,*}$ are local solutions to problem (21). In the following text, let $\boldsymbol{\beta}_{\mathbf{I}}$ indicates a vector whose elements are those of $\boldsymbol{\beta}$ with indices in $\mathbf{I}$. Let $\Delta = \mathbf{Z}^{i\,*} - \hat{\mathbf{Z}}^i$, $\tilde{\Delta} = \dot{\mathbf{P}}(\mathbf{Z}^{i\,*}) - \dot{\mathbf{P}}(\hat{\mathbf{Z}}^i)$. By Lemma 1, we have

$$\|\mathbf{D}^\top \mathbf{D}\Delta + \tilde{\Delta}\|_2 = 0$$

It follows that

$$\Delta^\top \mathbf{D}^\top \mathbf{D}\Delta + \Delta^\top \tilde{\Delta} \leq \|\Delta\|_2 \|\mathbf{D}^\top \mathbf{D}\Delta + \tilde{\Delta}\|_2 = 0$$

Also, by the proof of Lemma 1, for $k \in \hat{\mathbf{S}}_i \cap \mathbf{S}_i^*$, since $(\mathbf{D}^\top \mathbf{D}\Delta)_k = 2\lambda \mathbb{1}_{\mathbf{Z}_k^{i\,*}\hat{\mathbf{z}}_k^i < 0} + 0 \mathbb{1}_{\mathbf{Z}_k^{i\,*}\hat{\mathbf{z}}_k^i > 0}$ we have $\tilde{\Delta}_k = -(\mathbf{D}^\top \mathbf{D}\Delta)_k$. We now present another property on any nonconvex function $P$ using the degree of nonconvexity in Definition 3: $\theta(t, \kappa) := \sup_s \{-\mathrm{sgn}(s - t)(\dot{P}(s; b) - \dot{P}(t; b)) - \kappa|s - t|\}$ on the regularizer $\mathbf{P}$. For any $s, t \in \mathbb{R}$, we have

$$-\mathrm{sgn}(s - t)\big(\dot{P}(s; b) - \dot{P}(t; b)\big) - \kappa|s - t| \leq \theta(t, \kappa)$$

by the definition of $\theta$. It follows that

$$\theta(t, \kappa)|s - t| \geq -(s - t)\big(\dot{P}(s; b) - \dot{P}(t; b)\big) - \kappa(s - t)^2$$
$$-(s - t)\big(\dot{P}(s; b) - \dot{P}(t; b)\big) \leq \theta(t, \kappa)|s - t| + \kappa(s - t)^2 \tag{28}$$

Applying (28) with $P = P_k$ for $k = 1, \ldots, p$, we have

$$\Delta^\top \mathbf{D}^\top \mathbf{D}\Delta \leq -\Delta^\top \tilde{\Delta} = -\Delta_{\hat{\mathbf{S}}_i}^\top \tilde{\Delta}_{\hat{\mathbf{S}}_i} - \Delta_{\hat{\mathbf{S}}_i \cap \mathbf{S}_i^*}^\top \tilde{\Delta}_{\hat{\mathbf{S}}_i \cap \mathbf{S}_i^*}$$
$$\leq |\mathbf{Z}_{\hat{\mathbf{S}}_i}^{i\,*} - \hat{\mathbf{Z}}_{\hat{\mathbf{S}}_i}^i|^\top \theta(\hat{\mathbf{Z}}_{\hat{\mathbf{S}}_i}^i, \kappa) + \kappa\|\mathbf{Z}_{\hat{\mathbf{S}}_i}^{i\,*} - \hat{\mathbf{Z}}_{\hat{\mathbf{S}}_i}^i\|_2^2 + \|\Delta_{\hat{\mathbf{S}}_i \cap \mathbf{S}_i^*}\|_2 \|\tilde{\Delta}_{\hat{\mathbf{S}}_i \cap \mathbf{S}_i^*}\|_2$$
$$\leq \|\theta(\hat{\mathbf{Z}}_{\hat{\mathbf{S}}_i}^i, \kappa)\|_2 \|\mathbf{Z}_{\hat{\mathbf{S}}_i}^{i\,*} - \hat{\mathbf{Z}}_{\hat{\mathbf{S}}_i}^i\|_2 + \kappa\|\mathbf{Z}_{\hat{\mathbf{S}}}^i * - \hat{\mathbf{Z}}_{\hat{\mathbf{S}}}^i\|_2^2 + \|\Delta\|_2 \|\tilde{\Delta}_{\hat{\mathbf{S}}_i \cap \mathbf{S}_i^*}\|_2$$
$$\leq \|\theta(\hat{\mathbf{Z}}_{\hat{\mathbf{S}}_i}^i, \kappa)\|_2 \|\Delta\|_2 + \kappa\|\Delta\|_2^2 + \|\Delta\|_2 \|\tilde{\Delta}_{\hat{\mathbf{S}}_i \cap \mathbf{S}_i^*}\|_2 \tag{29}$$

On the other hand, $\Delta^\top \mathbf{D}^\top \mathbf{D}\Delta \geq \kappa_0^2 \|\Delta\|_2^2$. It follows from (29) that

$$\kappa_0^2 \|\Delta\|_2^2 \leq \|\theta(\hat{\mathbf{Z}}_{\hat{\mathbf{S}}_i}^i, \kappa)\|_2 \|\Delta\|_2 + \kappa\|\Delta\|_2^2 + \|\Delta\|_2 \|\tilde{\Delta}_{\hat{\mathbf{S}}_i \cap \mathbf{S}_i^*}\|_2$$

When $\|\Delta\|_2 \neq 0$, we have

$$\kappa_0^2 \|\Delta\|_2 \leq \|\theta(\hat{\mathbf{Z}}_{\hat{\mathbf{S}}_i}^i, \kappa)\|_2 + \kappa\|\Delta\|_2 + \|\tilde{\Delta}_{\hat{\mathbf{S}}_i \cap \mathbf{S}_i^*}\|_2$$
$$\Rightarrow \|\Delta\|_2 \leq \frac{\|\theta(\hat{\mathbf{Z}}_{\hat{\mathbf{S}}_i}^i, \kappa)\|_2 + \|\tilde{\Delta}_{\hat{\mathbf{S}}_i \cap \mathbf{S}_i^*}\|_2}{\kappa_0^2 - \kappa} \tag{30}$$

According to the definition of $\theta$, it can be verified that $\theta(t, \kappa) = \max\{0, \frac{\gamma \mathbf{G}_{ki}^{\mathbf{A}}}{b} - \kappa|t - b|\}$ for $|t| > b$, and $\theta(0+, \kappa) = \max\{0, \frac{\gamma \mathbf{G}_{ki}^{\mathbf{A}}}{b} - \kappa b\}$. Therefore,

$$\|\theta(\hat{\mathbf{Z}}_{\hat{\mathbf{S}}_i}^i, \kappa)\|_2 = \Big( \sum_{k \in \tilde{\mathbf{S}}_i \cap \hat{\mathbf{S}}_i} (\max\{0, \frac{\gamma \mathbf{G}_{ki}^{\mathbf{A}}}{b} - \kappa|\hat{\mathbf{Z}}_{ki} - b|\})^2 +$$
$$\sum_{k \in \tilde{\mathbf{S}}_i \setminus \hat{\mathbf{S}}_i} (\max\{0, \frac{\gamma \mathbf{G}_{ki}^{\mathbf{A}}}{b} - \kappa b\})^2 \Big)^{\frac{1}{2}} \tag{31}$$

Therefore,

$$\|\Delta\|_2 \leq \frac{1}{\kappa_0^2 - \kappa} \Big( \Big( \sum_{k \in \tilde{\mathbf{S}}_i \cap \hat{\mathbf{S}}_i} (\max\{0, \frac{\gamma \mathbf{G}_{ki}^{\mathbf{A}}}{b} - \kappa|\hat{\mathbf{Z}}_{ki} - b|\})^2 +$$
$$\sum_{k \in \tilde{\mathbf{S}}_i \setminus \hat{\mathbf{S}}_i} (\max\{0, \frac{\gamma \mathbf{G}_{ki}^{\mathbf{A}}}{b} - \kappa b\})^2 \Big)^{\frac{1}{2}} + \|\tilde{\Delta}_{\hat{\mathbf{S}}_i \cap \mathbf{S}_i^*}\|_2 \Big) \tag{32}$$

where $\tilde{\Delta}_k = -(\mathbf{D}^\top \mathbf{D}\Delta)_k = -2\lambda \mathbb{1}_{\mathbf{Z}_k^{i\,*}\hat{\mathbf{z}}_k^i < 0} - 0 \mathbb{1}_{\mathbf{Z}_k^{i\,*}\hat{\mathbf{z}}_k^i > 0}$ for $k \in \hat{\mathbf{S}}_i \cap \mathbf{S}_i^*$. This proves the result of this theorem.  □

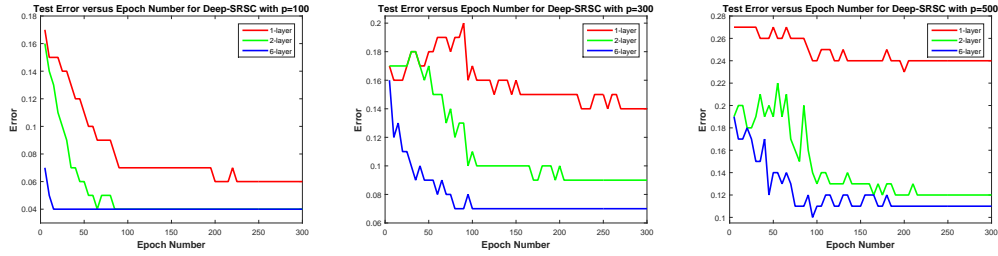

Figure 5: Test error of Deep-SRSC with dictionary size $p = 100$, $p = 300$, and $p = 500$

Table 5: Clustering Results on COIL-100 data with different dictionary size $p$

| Dictionary Size | Measure | KM | SC | Sparse Coding | $\ell^2$-RSC | SRSC |
|---|---|---|---|---|---|---|
| $p = 100$ | AC | 0.5221 | 0.2372 | 0.7010 | 0.7010 | **0.7344** |
| | NMI | 0.7633 | 0.5410 | 0.8834 | 0.8834 | **0.8950** |
| $p = 300$ | AC | - | - | 0.6979 | 0.6979 | **0.7267** |
| | NMI | - | - | 0.8837 | 0.8837 | **0.8876** |
| $p = 500$ | AC | - | - | 0.6979 | 0.6979 | **0.7117** |
| | NMI | - | - | 0.8839 | 0.8839 | **0.8856** |

Table 6: Parameter sensitivity with respect to $\gamma$ and $K$ on USPS data set

| Varying $\gamma$ with default $K = 3$ | Measure | KM | SC | Sparse Coding | $\ell^2$-RSC | SRSC |
|---|---|---|---|---|---|---|
| $\gamma = 0.1$ | AC | 0.6878 | 0.4041 | 0.8178 | **0.8287** | 0.8229 |
| | NMI | 0.6312 | 0.4765 | 0.8321 | **0.8398** | 0.8370 |
| $\gamma = 0.2$ | AC | - | - | 0.8178 | **0.8287** | 0.8261 |
| | NMI | - | - | 0.8321 | 0.8398 | **0.8439** |
| $\gamma = 0.3$ | AC | - | - | 0.8178 | **0.8287** | 0.8251 |
| | NMI | - | - | 0.8321 | 0.8398 | **0.8441** |
| $\gamma = 0.4$ | AC | - | - | 0.8178 | **0.8287** | 0.8258 |
| | NMI | - | - | 0.8321 | 0.8398 | **0.8455** |
| $\gamma = 0.5$ | AC | - | - | 0.8178 | 0.8287 | **0.8293** |
| | NMI | - | - | 0.8321 | 0.8398 | **0.8471** |
| $\gamma = 0.6$ | AC | - | - | 0.8178 | **0.8287** | 0.8273 |
| | NMI | - | - | 0.8321 | 0.8398 | **0.8481** |
| $\gamma = 0.7$ | AC | - | - | 0.8178 | **0.8287** | 0.8279 |
| | NMI | - | - | 0.8321 | 0.8398 | **0.8489** |
| $\gamma = 0.8$ | AC | - | - | 0.8178 | **0.8287** | 0.8282 |
| | NMI | - | - | 0.8321 | 0.8398 | **0.8479** |
| Varying $K$ with default $\gamma = 0.5$ | Measure | KM | SC | Sparse Coding | $\ell^2$-RSC | SRSC |
| $K = 3$ | AC | - | - | 0.8178 | 0.8287 | **0.8293** |
| | NMI | - | - | 0.8321 | 0.8398 | **0.8471** |
| $K = 4$ | AC | - | - | 0.8178 | **0.8287** | 0.8216 |
| | NMI | - | - | 0.8321 | 0.8398 | **0.8487** |
| $K = 5$ | AC | - | - | 0.8178 | **0.8287** | 0.8243 |
| | NMI | - | - | 0.8321 | 0.8398 | **0.8535** |
| $K = 6$ | AC | - | - | 0.8178 | 0.8287 | **0.8462** |
| | NMI | - | - | 0.8321 | **0.8398** | 0.7995 |

Table 7: Clustering results on the test set of MNIST Data

| Measure | KM | SC | Sparse Coding | $\ell^2$-RSC | SRSC | 6-Layer Deep-SRSC |
|---|---|---|---|---|---|---|
| AC | 0.5606 | 0.3452 | 0.6039 | 0.6312 | 0.7513 | **0.7621** |
| NMI | 0.5382 | 0.4129 | 0.6210 | 0.6749 | 0.7378 | **0.7537** |

Table 8: Clustering results on the test set of CIFAR-10 Data

| Measure | KM | SC | Sparse Coding | $\ell^2$-RSC | SRSC | 6-Layer Deep-SRSC |
|---|---|---|---|---|---|---|
| AC | 0.4548 | 0.4220 | 0.4113 | 0.4539 | **0.4625** | 0.4574 |
| NMI | 0.3655 | 0.3133 | 0.3335 | 0.3670 | **0.3869** | 0.3516 |

## MORE EXPERIMENTAL RESULTS

The test error of Deep-SRSC with different dictionary size, corresponding to Figure 4 showing the training error, is illustrated in Figure 5. We vary the dictionary size and show the clustering results on COIL-100 data set in Table 5, and we can see that SRSC always achieves highest accuracy and NMI with different dictionary size.

In addition, we investigate the parameter sensitivity of SRSC, and show in Table 6 the performance change while varying $\gamma$, the weight for the support regularization term, and $K$, the number of nearest neighbors when building the KNN graph for the support regularization term, on the USPS data set. It can be observed that the performance of SRSC is stable over a relatively large range of $\lambda$ and $K$. SRSC often has the highest NMI while maintaining a very competitive accuracy.

## DEEP-SRSC WITH THE SECOND TEST SETTING (REFERRING TO THE TRAINING DATA)

We demonstrate the performance of SRSC and Deep-SRSC with the second test setting (referring to the training data) on clustering and semi-supervised learning. The ground truth code of the each test data point is computed by performing the PGD-style iterative method to solve the problem (8) where $\mathbf{x}_i$ is the test point, $\mathbf{D}$ is $\mathbf{D}_{sr}$ obtained from the training data as in Section 5.2, $\mathbf{A}$ is the adjacency matrix of the KNN graph over the test point and the training data. Table 9 shows the prediction error of Deep-SRSC for different dictionary size $p$ and different number of layers on the USPS data, which is comparable to the case of the first test setting in Table 3.

Two more data sets are used in this subsection, i.e. MNIST for hand-written digit recognition and CIFAR-10 for image recognition. MNIST is comprised of 60000 training images and 10000 test images of ten digits from 0 to 9, and each image is of size $28 \times 28$ and represented as a 784-dimensional vector. CIFAR-10 consists of 50000 training images and 10000 testing images in 10 classes, and each image is a color one of size $32 \times 32$. Using the second test setting, Deep-SRSC is trained on the training set, and the codes of the test set predicted by 6-layer Deep-SRSC are used to perform clustering on the test set for MNIST and CIFAR-10 data, with comparison to other sparse coding based methods. The clustering results are shown in Table 7 and 8 respectively with dictionary size $p = 300$. We observe that SRSC and Deep-SRSC always achieve the best performance compared to other competing methods. We employ the fast deep neural network named CNN-F (Chatfield et al., 2014) trained on the ILSVRC 2012 data to extract the 4096-dimensional feature vector for each image in the CIFAR-10 data, and all the clustering methods are performed on the extracted features. In addition to the coordinate descent method employed in Section 2.1.2 and Algorithm 1 for the optimization of the sparse codes in SRSC, we further conduct the empirical study showing that the parallel coordinate descent method, which updates the coordinates in parallel for improved efficiency and fits the needs of large-scale data optimization, leads to almost the same results as the coordinate descent method on the CIFAR-10 data. Instead of optimization with respect to the sparse code of a single data point in the coordinate descent method, the parallel coordinate descent method updates the sparse codes of $P$ data points in parallel using the same rule as that in the coordinate descent method in Section 2.1.2 and Algorithm 1. While the parallel coordinate descent method is originally designed for convex problems (Bradley et al., 2011; Richtárik & Takáč, 2016), it demonstrates almost the same empirical performance as the coordinate descent method for the clustering task on the test set of the CIFAR-10 data, with the accuracy of 0.4622 and NMI of 0.3864. $P$-parallel coordinate descent leads to $P$ times speedup compared to the coordinate descent method. We choose $P = 10$ and the codes of the training data of CIFAR-10 are learned by the parallel coordinate descent method, and note that the optimization of the codes of the test data are inherently parallelizable due to the nature of the second test setting studied in this subsection.

Moreover, Table 10 shows the prediction error of Deep-SRSC on the MNIST data and the CIFAR-10 data. It can be observed again that deeper Deep-SRSC network leads to smaller prediction error.

We also show the application to semi-supervised learning via label propagation (Zhu et al., 2003), a widely used semi-supervised learning method. Given the data $\{\mathbf{x}_1, \mathbf{x}_2, \ldots, \mathbf{x}_l, \mathbf{x}_{l+1}, \ldots, \mathbf{x}_n\} \subset \mathbb{R}^d$, the first $l$ points $\{\mathbf{x}_1, \mathbf{x}_2, \ldots, \mathbf{x}_l\}$ are labeled and named the training data, and the remaining $n - l$ points form the test data for semi-supervised learning. Semi-supervised learning by label propagation aims to predict the labels of the test data by encouraging local smoothness of the labels in accordance with the similarity matrix over the entire data. The performance of label propagation depends on the similarity matrix. For each sparse coding based method, the similarity matrix $\mathbf{Y}$ over the entire data is built by the support similarity introduced in Section 5.1: $\mathbf{Y}_{ij} = \mathbf{A}_{ij} \mathbf{q}_{\hat{\mathbf{Z}}^i}^\top \mathbf{q}_{\hat{\mathbf{Z}}^j}$, and $\mathbf{Z}^i$ is the code of data point $\mathbf{x}_i$ for different sparse coding methods including the 6-layer Deep-SRSC with the second test setting. Label propagation is performed on the similarity matrix $\mathbf{Y}$ to obtain the labels of the test data, and the error rate is reported. Note that in the experiment of semi-supervised learning by label propagation, the codes of the test data of each data set are obtained first (e.g. the 10000 test images in the MNIST data). If $\mathbf{x}_i$ belongs to the test data of a data set, its code is obtained by performing the the corresponding sparse coding optimization with the dictionary learned on the training data of that data set; for SRSC and Deep-SRSC, such optimization also has the KNN graph over the test point $\mathbf{x}_i$ and the training data as input. With the codes of all the data, the similarity matrix $\mathbf{Y}$ over the entire data is constructed. Then, a randomly sampled subset of each class is labeled as the training data, with the other data serving as the test data for semi-supervised learning.

The semi-supervised learning results of our methods are compared to that of the Gaussian kernel graph (Gaussian), i.e. the KNN graph with the edge weight set by the Gaussian kernel; Sparse Coding (SC) and $\ell^2$-RSC; and manifold based similarity adaptation (MBS) by Karasuyama & Mamitsuka (2013), one of the state-of-the-art semi-supervised learning methods based on label propagation. MBS learns the manifold aligned edge similarity by local reconstruction for label propagation.

The comparison results of semi-supervised learning by label propagation on the USPS data and the MNIST data are shown in Figure 6 and 7, which illustrate the error rate of label propagation with respect to different number of labeled data points in each class. We can observe from Figure 6 that SRSC and Deep-SRSC with the second test setting lead to superior results on the application to semi-supervised learning, and the performance of SRSC and Deep-SRSC is always the best with respect to different dictionary size. It can also be observed from Figure 6 and 7 that SRSC and Deep-SRSC have very similar performance, revealing the good quality of the fast approximation by Deep-SRSC on the semi-supervised learning task. Furthermore, SRSC and Deep-SRSC significantly outperform other baseline methods with the small number of labeled data points in each class, due to the captured locally linear manifold structure.

Table 9: Prediction error (average squared error between the predicted codes and the ground truth codes) of Deep-SRSC with different depth and different dictionary size on the test set of the USPS data, using the second test setting

| Dictionary Size | 1-layer | 2-layer | 6-layer |
|---|---|---|---|
| $p = 100$ | 0.05 | 0.03 | 0.03 |
| $p = 300$ | 0.12 | 0.08 | 0.06 |
| $p = 500$ | 0.27 | 0.11 | 0.09 |

Table 10: Prediction error (average squared error between the predicted codes and the ground truth codes) of Deep-SRSC with different depth on the test set of the MNIST data and CIFAR-10 data, using the second test setting

| Data Set | 1-layer | 2-layer | 6-layer |
|---|---|---|---|
| MNIST | 0.15 | 0.12 | 0.10 |
| CIFAR-10 | 0.16 | 0.13 | 0.13 |

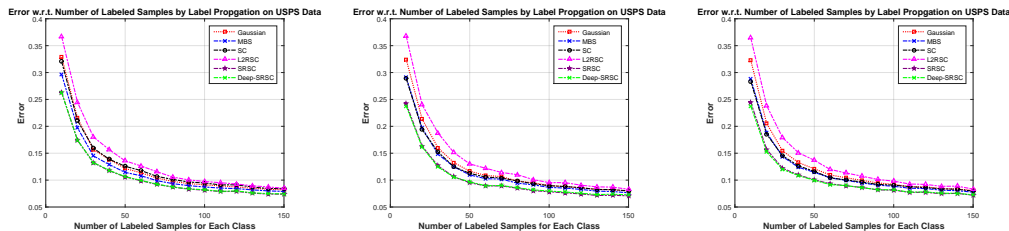

Figure 6: Error rate of semi-supervised learning by label propagation on the USPS data, with dictionary size $p = 100$, $p = 300$, and $p = 500$

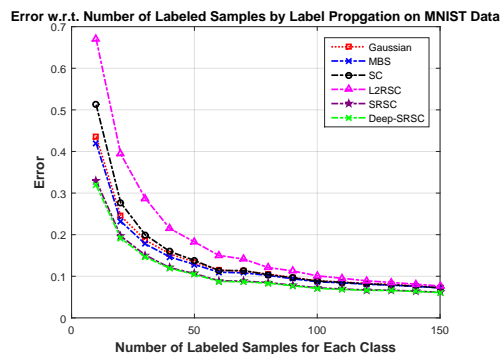

Figure 7: Error rate of semi-supervised learning by label propagation on the MNIST data with dictionary size $p = 300$

