# Peer review of "Support Regularized Sparse Coding and Its Fast Encoder"

_ICLR 2017 — accepted_

[Official Review · AnonReviewer2 · rating 7 · confidence 4 · 17 Dec 2016]

I'd like to thank the authors for their detailed response to my questions.

The paper proposes a support regularized version of sparse coding that takes into account the underlying manifold structure of the data. For this purpose, the authors augment the classic sparse coding loss with a term that encourages near by points to have similar active set. Convergence guarantees for the optimization procedure are presented. Experimental evaluation on clustering and semi-supervised learning shows the benefits of the proposed approach.

The paper is well written and a nice read. The most relevant contribution of this work is to including (and optimizing) the regularization function, and not an approximation or surrogate. The authors derive a a PGD-styple iterative method and present convergence analysis for it. 

Thanks for the clarifications regarding the assumptions used in Section 3. It would be nice to include some of that in the manuscript.

The authors also propose a fast encoding scheme for their proposed method. 
The authors included a new experiment in semi-supervised consists of a very interesting use (of the method and the fast approximation). While this is an interesting addition, I think that using fast encoders is not particularly novel or the main part of the work. "Converting" iterative optimization algorithms into feed-forward nets for accelerating the inference process has been done in the past (several times with quite similar problems). Is natural that this can be done, and not very surprising. Maybe would be interesting to evaluate how important is to have an architecture matching the optimization algorithm, compared to a generic network (though some of this analysis has also been performed in the past).

[Official Review · AnonReviewer3 · rating 6 · confidence 3 · 19 Dec 2016 (modified: 25 Jan 2017)]
**Interesting, but weak experimental results**

In this paper the authors propose a method to explicitly regularize sparse coding to encode neighbouring datapoints with similar sets of atoms from the dictionary by clustering training examples with KNN in input space. The resulting algorithm is relatively complex and computationally relatively expensive, but the authors provide detailed derivations and use arguments from proximal gradient descent methods to prove convergence (I did not follow all the derivations, only some). In general the paper is well written and the authors explain the motivation behind the algorithms design in detail. 

In the abstract the authors mention “extensive experimental results …”, but I find the experiments not very convincing: With experiments on the USPS handwritten digits dataset (why not MNIST?), COIL-20 and COIL-100 and UCI, the datasets are all relatively small and the algorithm is run with dictionary sizes between p=100 to p=500. This seems surprising because the authors state that they implemented SRSC in “CUDA C++ with extreme efficiency” (page 10). But more importantly, I find it hard to interpret and compare the results: The paper reports accuracy and and normalized mutual information for a image retrieval / clustering task where the proposed SRSC is used as a feature extractor. The improvements relative to standard Sparse Coding seem very small  (often < 1% in terms of NMI; it looks more promising in terms of accuracy) and if I understand the description on page 11 correctly, than the test set was used to select some hyperparameters (the best similarity measure for clustering step)? There is no comparisons to other baselines / state of the art image clustering methods.
Besides of providing features for a small scale image clustering system, are there maybe ways to more directly evaluate the properties and qualities of  a sparse coding approach? E.g. reconstruction error / sparsity; maybe even denoising performance?
 
In summary, I think in it current form the paper lacks the evaluation and experimental results for an ICLR publication. Intuitively, I agree with the authors that the proposed regularization is an interesting direction, but I don’t see experiments that directly show that the regularization has the desired effect; and the improvements in the clustering task where SRSC is used as a feature extractor are very modest.

[Official Review · AnonReviewer1 · rating 7 · confidence 4 · 21 Dec 2016]
**Support Regularized Sparse Coding and Its Fast Encoder**

The work proposes to use the geometry of data (that is considered to be known a priori) in order to have more consistent sparse coding. Namely, two data samples that are similar or neighbours, should have a sparse code that is similar (in terms of support). The general idea is not unique, but it is an interesting one (if one admits that the adjacency matrix A is known a priori), and the novelty mostly lies on the definition of the regularisation term that is an l1-norm (while other techniques would mostly use l2 regularisation).

Based on this idea, the authors develop a new SRSC algorithm, which is analysed in detail and shown to perform better than its competitors based on l2 sparse coding regularisation and other schemes in terms of clustering performance.

Inspired by LISTA, the authors then propose an approximate solution to the SRSC problem, called Deep-SRSC, that acts as a sort of fast encoder. Here too, the idea is interesting and seems to be quite efficient from experiments on USPS data, even if the framework seems to be strongly inspired from LISTA. That scheme should however be better motivated, by the limitations of SRSC that should be presented more clearly. 

Overall, the paper is well written, and pretty complete. It is not extremely original in its main ideas though, but the actual algorithm and implementation seem new and effective.

[Final Decision · Program Chairs · 06 Feb 2017]
**ICLR committee final decision**

Adding a manifold regularizer to a learning objective function is certainly not a new direction. The paper argues that using a support based regularizer is superior to using a standard graph Laplacian regularizer (which has been explored before), although this argument is not developed particularly rigorously and dominantly has to fall back on empirical evidence. The main contribution of the paper appears to be theoretical justification of an alternating optimization scheme for minimizing the resulting objective function (yet the optimization aspects of dealing with a sparse support regularizer are somewhat orthogonal to the current context). The empirical results are not very convincing since the dictionary size is relatively large compared to the dataset size; the gains with respect to l2 manifold regularizer are not consistent; and the gains using deep architectures to directly predict sparse codes are also modest and somewhat inconsistent. These points aside, the reviewers are overall enthusiastic about the paper and find it to be well written and complete.